# Signal Processing for Implicit Neural Representations

**Dejia Xu**[*]
dejia@utexas.edu

**Peihao Wang**[*]
peihaowang@utexas.edu

**Yifan Jiang**
yifanjiang97@utexas.edu

**Zhiwen Fan**
zhiwenfan@utexas.edu

**Zhangyang Wang**
atlaswang@utexas.edu

The University of Texas at Austin
https://vita-group.github.io/INSP/

## Abstract

Implicit Neural Representations (INRs) encoding continuous multi-media data via multi-layer perceptrons has shown undebatable promise in various computer vision tasks. Despite many successful applications, editing and processing an INR remains intractable as signals are represented by latent parameters of a neural network. Existing works manipulate such continuous representations via processing on their discretized instance, which breaks down the compactness and continuous nature of INR. In this work, we present a pilot study on the question: *how to directly modify an INR without explicit decoding?* We answer this question by proposing an implicit neural signal processing network, dubbed **INSP-Net**, via differential operators on INR. Our key insight is that spatial gradients of neural networks can be computed analytically and are invariant to translation, while mathematically we show that any continuous convolution filter can be uniformly approximated by a linear combination of high-order differential operators. With these two knobs, INSP-Net instantiates the signal processing operator as a weighted composition of computational graphs corresponding to the high-order derivatives of INRs, where the weighting parameters can be data-driven learned. Based on our proposed INSP-Net, we further build the first Convolutional Neural Network (CNN) that implicitly runs on INRs, named **INSP-ConvNet**. Our experiments validate the expressiveness of INSP-Net and INSP-ConvNet in fitting low-level image and geometry processing kernels (e.g. blurring, deblurring, denoising, inpainting, and smoothening) as well as for high-level tasks on implicit fields such as image classification.

## 1 Introduction

The idea that our visual world can be represented continuously has attracted increasing popularity in the field of implicit neural representations (INR). Also known as coordinate-based neural representations, INRs learn to encode a coordinate-to-value mapping for continuous multi-media data. Instead of storing the discrete signal values in a grid of pixels or voxels, INRs represent discrete data as samples of a continuous manifold. Using multi-layer perceptrons, INRs bring practical benefits to various computer vision applications, such as image and video compression [1, 2, 3], 3D shape representation [4, 5, 6, 7, 8, 9, 10, 11], inverse problems [12, 2, 13, 14], and generative models [15, 16, 17, 18, 19, 20, 21, 22].

Despite their recent success, INRs are not yet amenable to flexible editing and processing as the standard images could do. The encoded coordinate-to-value mapping is too complex to comprehend and the parameters stored in multi-layer perceptrons (MLPs) remains less explored. One direction of existing approaches enables editing on INRs by training them with conditional input. For example,

---

[*]Equal Contribution.

36th Conference on Neural Information Processing Systems (NeurIPS 2022).

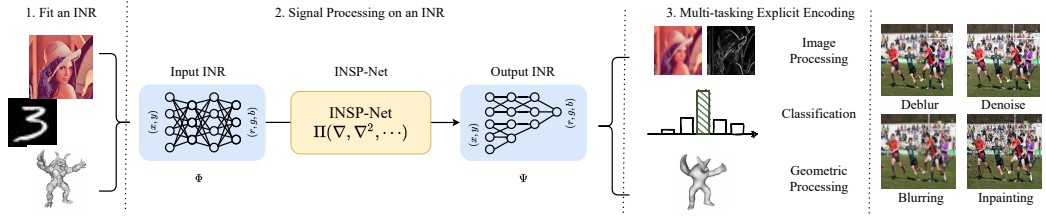

Figure 1: An illustration of implicit neural signal processing. Given an INR representing digital signals, our INSP-Net is capable of direct signal processing without needing to explicitly decode it. Our model first constructs derivative computation graphs of the original INR and then generates a linear combination of them into a new INR. It can be later decoded into discretized forms such as image pixels. The framework is capable of fitting low-level image processing kernels as well as performing high-level processing such as image classification.

[23, 24, 25, 20, 21, 26] utilize conditional codes to indicate different characteristics of the scene including shape and color. Another main direction benefits from existing image editing techniques and operates on discretized instances of continuous INRs such as pixels or voxels. However, such solutions break down the continuous characteristic of INR due to the prerequisite of decoding and discretizing before editing and processing.

In this paper, we conduct the first pilot study on the question: *how to generally modify an INR without explicit decoding?* The major challenge is that one cannot directly interpret what the parameters in an INR stand for, not to mention editing them correctly. Our key motivation is that spatial gradients can be served as a favorable tool to tackle this problem as they can be computed analytically, and possess desirable invariant properties. Theoretically, we prove that any continuous convolution filter can be uniformly approximated by a linear combination of high-order differential operators. Based on the above two rationales, we propose an Implicit Neural Signal Processing Network, dubbed **INSP-Net**, which processes INR utilizing high-order differential operators. The proposed INSP-Net is composed of an inception fusion block connecting computational graphs corresponding to derivatives of INRs. The weights in the branchy part are loaded from the INR being processed, while the weights in the fusion block are parameters of the operator, which can be either hand-crafted or learned by the data-driven algorithm. Even though we are not able to perform surgery on neural network parameters, we can implicitly process them by retrofitting their architecture and reorganizing the spatial gradients.

We further extend our framework to build the first Convolutional Neural Network (CNN) operating directly on INRs, dubbed INSP-ConvNet. Each layer of INSP-ConvNet is constructed by linearly combining the derivative computational graphs of the former layers. Nonlinear activation and normalization are naturally supported as they are element-wise functions. Data augmentation can be also implemented by augmenting the input coordinates of INRs. Under this pipeline (shown in Fig. 1), we demonstrate the expressiveness of our INSP-Net framework in fitting low-level image processing kernels including edge detection, blurring, deblurring, denoising, and image inpainting. We also successfully apply our INSP-ConvNet to high-level tasks on implicit fields such as classification.

Our main contributions can be summarized as follows:

- We propose a novel signal processing framework, dubbed **INSP-Net**, that operates on INRs analytically and continuously by closed-form high-order differential operators[2]. Repeatedly cascading the computational paradigm of INSP-Net, we also build a convolutional network, called INSP-ConvNet, which directly runs on implicit fields for high-level tasks.

- We illustrate the advantage of adopting differential operators by revealing their inherent group invariance. Furthermore, we rigorously prove that the convolution operator in the continuous regime can be uniformly approximated by a linear combination of the gradients.

- Extensive experiments demonstrate the effectiveness of our approach in both low-level processing (e.g. edge detection, blurring, deblurring, denoising, image inpainting, and smoothening) and high-level processing such as image classification.

---

[2]By saying "closed-form", we mean the computation follows from an analytical mathematical expression.

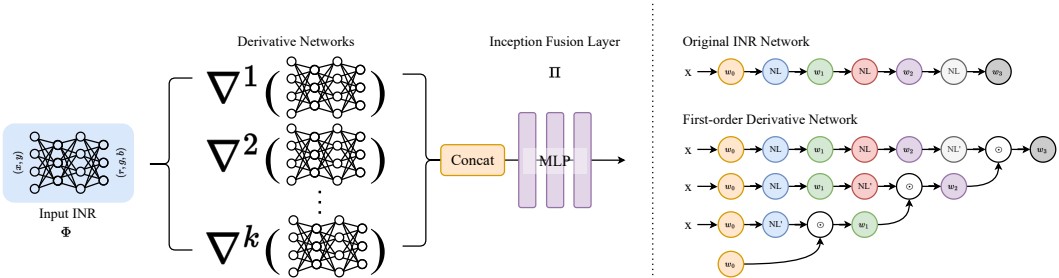

Figure 2: The left image provides an overview of our INSP-Net framework. Each layer combines the high-order derivative computational graphs of the original INR network. The right image illustrates the weight sharing scheme in calculating the derivative sub-networks.

## 2 Preliminaries: Implicit Neural Representation

Implicit Neural Representation (INR) parameterizes continuous multi-media signals or vector fields with neural networks. Formally, we consider an INR as a continuous function $\Phi : \mathbb{R}^m \rightarrow \mathbb{R}$ that maps low-dimension spatial/temporal coordinates to the value space[3]. For example, to represent 2D image signals, the domain of $\Phi$ is $(x, y)$ spatial coordinates, and the range of $\Phi$ are the pixel intensities. The typical use of INR is to solve a feasibility problem where $\Phi$ is sought to satisfy a set of $N$ constraints $\{\mathcal{C}(\Phi, a_j | \Omega_j)\}_{j=1}^N$, where $\mathcal{C}$ is a functional that relates function $\Phi$ to some observable quantities $a_j$ evaluating over a measurable domain $\Omega_j \subseteq \mathbb{R}^m$ . This problem can be cast into an optimization problem that minimizes deviations from each of the constraints:

$$\Phi^* = \arg \min_{\Phi} \sum_{j=1}^N \|\mathcal{C}(\Phi, a_j | \Omega_j)\|_2. \tag{1}$$

For instance, we can let $\mathcal{C} = \Phi(\boldsymbol{x}_j) - a_j$ with $\Omega_j = \{\boldsymbol{x}_j\}$, then our objective boils down to a point-to-point supervision which memorizes a signal into $\Phi$ [27]. When functional $\mathcal{C}$ is a combination of differential operators taking values in a point set, i.e., $\mathcal{C}(a(\boldsymbol{x}), \Phi(\boldsymbol{x}), \nabla\Phi(\boldsymbol{x}), \cdots), \forall \boldsymbol{x} \in \Omega_j$, Eq. 1 is objective to solving a bunch of differential equations [28, 7, 29]. Note that in this paper, without particular specification, the gradients are all computed with respect to the input coordinate $\boldsymbol{x}$. $\mathcal{C}$ can also form an integral equation system over some intervals $\Omega_j$ [12]. In practice of computer vision, we reconstruct a signal by capturing sparse observations $\mathcal{D} = \{(\Omega_j, a_j)\}_{j=1}^N$ from unknown function $\Phi$, and dynamically sampling a mini-batch from $\mathcal{D}$ to minimize Eq. 1 to obtain a feasible $\Phi$.

A handy parameterization of function $\Phi$ is a fully-connected neural network, which enables solving Eq. 1 via gradient descent through a differentiable $\mathcal{C}$. Common INR networks consist of pure Multi-Layer Perceptrons (MLP) with periodic activation functions. Fourier Feature Mapping (FFM) [27] places a sinusoidal transformation before the MLP, while Sinusoidal Representation Network (SIREN) [28] replaces every piece-wise linear activation with a sinusoidal function. Below we give a unified formulation of INR networks:

$$\Phi(\boldsymbol{x}) = \boldsymbol{W}_n(\phi_{n-1} \circ \phi_{n-2} \circ \cdots \circ \phi_1)(\boldsymbol{x}), \quad \phi_i(\boldsymbol{x}) = \sigma_i(\boldsymbol{W}_i\boldsymbol{x} + \boldsymbol{b}_i), \tag{2}$$

where $\boldsymbol{W}_i \in \mathbb{R}^{d_{i-1} \times d_i}, \boldsymbol{b}_i \in \mathbb{R}^{d_i}$ are the weight matrix and bias of the $i$-th layer, respectively, $n$ is the number of layers, and $\sigma_i(\cdot)$ is an element-wise nonlinear activation function. For FFM architecture, $\sigma_i = \sin(\cdot)$ when $i = 1$ denotes the positional encoding layer [12, 30] and otherwise $\sigma_i = \text{ReLU}(\cdot)$. For SIREN, $\sigma_i = \sin(\cdot)$ for every layer $i = 1, \cdots, n-1$.

## 3 Implicit Representation Processing via Differential Operators

Digital Signal Processing (DSP) techniques have been widely applied in computer vision tasks, such as image restoration [31], signal enhancement [32] and geometric processing [33]. Even modern deep

---

[3]Without loss of generality, here we simplify $\Phi$ to be a scalar field, i.e., the range of $\Phi$ is one-dimensional.

learning models are consisting of the most basic signal processing operators. Suppose we already acquire an Implicit Neural Representation (INR) $\Phi : \mathbb{R}^m \to \mathbb{R}$, now we are interested in whether we can run a signal processing program on the implicitly represented signals. One straightforward solution is to rasterize the implicit field with a 2D/3D lattice and run a typical kernel on the pixel/voxel grids. However, this decoding strategy produces a finite resolution and discretizes signals, which is memory inefficient and unfriendly to modeling fine details. In this section, we introduce a computation paradigm that can process an INR analytically with spatial/temporal derivatives. We show that our proposed method serves as a universal operator that can represent any continuous convolutional kernels.

## 3.1 Computational Paradigm

It has not escaped our notice that spatial/temporal gradients on INRs $\nabla^k \Phi$ can be computed analytically due to the differentiable characteristics of neural networks. Inspired by this, we propose an Implicit Neural Signal Processing (INSP) framework that composes a class of closed-form operators for INRs using functional combinations of high-order derivatives.

We denote our proposed signal processing operator by $\mathcal{A}$ built upon high-order derivatives. Given an acquired INR $\Phi$, we denote the resultant INR processed by operator $\mathcal{A}$ as $\Psi = \mathcal{A}\Phi : \mathbb{R}^m \to \mathbb{R}$. To evaluate point $\boldsymbol{x} \in \mathbb{R}^m$ of processed INR, we propose the following computational paradigm:

$$\Psi(\boldsymbol{x}) := \mathcal{A}\Phi(\boldsymbol{x}) = \Pi \left( \Phi(\boldsymbol{x}), \nabla\Phi(\boldsymbol{x}), \nabla^2\Phi(\boldsymbol{x}), \cdots, \nabla^k\Phi(\boldsymbol{x}), \cdots \right), \tag{3}$$

where $\Pi : \mathbb{R}^M \to \mathbb{R}$ can be arbitrary continuous functions, which can be either handcrafted or learned from data. To learn an operator $\mathcal{A}$ from data, we represent $\Pi$ by Multi-Layer Perceptrons (MLP) with parameters $\boldsymbol{\theta}$. Here we can slightly abuse the notation of $\nabla^k$ to be a flattened vector of high-order derivatives *without multiplicity* since differential operators defined over continuous functions form a commutative ring. The input dimension of $\Pi$ depends on the highest order of used derivatives. Suppose we compute derivatives up to $K$-th order, then $M = \sum_{k=0}^{K} \binom{k+m-1}{k} = (K+1)\binom{K+m}{K+1}/m$, where $\binom{k+m-1}{k}$ is the number of distinctive $k$-th order differential operators [4]. Intuitively, directional derivatives encode (local) neighboring information, which can have similar effects of a convolution. As we will show in Sec. 3.2, $\Pi$ can construct both shift-invariant and rotation-invariant operators, which introduces favorable inductive bias to images and 3D geometry processing. More importantly, we rigorously prove that Eq. 3 is also a universal approximator of arbitrary convolutional operators.

We note that $\Psi(\boldsymbol{x})$ as a whole can also be regarded as a neural network. Recall the architecture of $\Phi(\boldsymbol{x})$ in Eq. 2, its $k$-th order derivative is another computational graph parameterized by $\boldsymbol{W}_i$ and $\boldsymbol{b}_i$ that maps $\boldsymbol{x}$ to $\nabla^k\Phi(\boldsymbol{x})$. For example, the first-order gradient will have the following form:

$$\nabla\Phi(\boldsymbol{x}) = \hat{\phi}_{n-1} \circ (\phi_{n-2} \circ \cdots \circ \phi_1)(\boldsymbol{x}) \odot \cdots \odot \hat{\phi}_2 \circ \phi_1(\boldsymbol{x}) \odot \boldsymbol{W}_1, \tag{4}$$

where $\hat{\phi}_i(\boldsymbol{y}) = \boldsymbol{W}_i^\top \sigma_{i-1}'(\boldsymbol{W}_{i-1}\boldsymbol{y} + \boldsymbol{b}_{k-1})$, and $\sigma_i'(\cdot)$ is the first-order derivative of $\sigma_i(\cdot)$. Since $\hat{\phi}_i$ shares the weights with $\phi_i$, $\nabla\Phi$ is represented by a closed-form computational network re-using the weights from $\Phi$, which we refer to as the first-order *derivative network*. The higher-order derivatives should induce the derivative network of similar forms. Therefore, the processed INR $\Psi$ will have an Inception-like architecture, namely, a multi-branch structure connecting the original INR network and weight-sharing derivative subnetworks followed by a fusion layer $\Pi$. We call the entire model ($\Psi = \mathcal{A}\Phi$ or Eq. 3) an *Implicit Neural Signal Processing Network* or an *INSP-Net*. Note that the only parameters of INSP-Net $\boldsymbol{\theta}$ are located at the last fusion layer, and can be trained in an end-to-end manner.

We illustrate an INSP-Net in Fig. 2 where the color indicates the weight-sharing scheme. A similar weight-sharing scheme is also adopted in AutoInt [34]. In practice, we employ auto-differentiation in PyTorch [35] to automatically create such derivatives networks and reassemble them parallelly to constitute the architecture of an INSP-Net. When inputting an INR, we load the weights of the INR to our model following the weight-sharing scheme, and then we obtain an INSP-Net, which implicitly and continuously represents the processed INR $\Psi(\boldsymbol{x})$. To effectively express high-order derivatives, we choose SIREN as the base model [28].

## 3.2 Theoretical Analysis

In this section, we provide a theoretical justification for the design of our INSP-Net. We will focus on discussing the latent invariance property and the expressive power of INSP-Net.

---

[4]This is equal to the number of monic monomials over $\mathbb{R}^m$ with degree $k$.

**Translation and Rotation Invariance.** Group invariance has been shown to be a favorable inductive bias for image [36], video [37], and geometry processing [38]. It has also been well-known that group invariance is an intrinsic property of Partial Differential Equations (PDEs) [39, 40]. Since our INSP-Net is built using differential operators, we are motivated to reveal its hidden invariance property to demonstrate its advantage in processing visual signals.

In this section, we only consider two transformation groups: translation group $\mathbb{T}(m)$ and the special orthogonal group $\mathbb{SO}(m)$ (a.k.a. rotation group). Elements $T_{\boldsymbol{v}} \in \mathbb{T}(m)$ in translation group shift the function $\Phi$ by some offset $\boldsymbol{v} \in \mathbb{R}^m$. The shifted function can be denoted as $\Phi \circ T_{\boldsymbol{v}}(\boldsymbol{x}) = \Phi(\boldsymbol{x} + \boldsymbol{v})$. Similarly, elements in rotation group perform a coordinate transformation on function $\Phi$ by a rotation matrix $\boldsymbol{R} \in \mathbb{SO}(m)$. The transformed function can be written as $\Phi \circ \boldsymbol{R}(\boldsymbol{x}) = \Phi(\boldsymbol{R}\boldsymbol{x})$. Group invariance means deforming the input space of a function first and then processing it via an operator is equivalent to directly applying the transformation to the processed function. For a more rigorous argument, $\mathcal{A}$ is said to be translation-invariant if $\forall T_{\boldsymbol{v}} \in \mathbb{T}(m)$, $\Psi(\boldsymbol{x} + \boldsymbol{v}) = \mathcal{A}[\Phi \circ T_{\boldsymbol{v}}](\boldsymbol{x})$. Likewise, $\mathcal{A}$ is rotation-invariant if $\forall \boldsymbol{R} \in \mathbb{SO}(m)$ we have $\Psi(\boldsymbol{R}\boldsymbol{x}) = \mathcal{A}[\Phi \circ \boldsymbol{R}](\boldsymbol{x})$. Below we provide Theorem 1 to characterize the invariance property for our model.

**Theorem 1.** *Given function $\Pi : \mathbb{R}^M \to \mathbb{R}$, the composed operator $\mathcal{A}$ (Eq. 3) can satisfy:*

1. *shift invariance for every $\Pi$.*

2. *rotation invariance if $\Pi$ has the form: $\Pi(\boldsymbol{y}) = f(\|\boldsymbol{y}\|_2)$ for some $f : \mathbb{R} \to \mathbb{R}$.*

We prove Theorem 1 in Appendix A. Our Theorem 1 implies that operator $\mathcal{A}$ is inherently shift-invariant. This is due to the shift-invariant intrinsics of differential operators as we show in the proof. Rotation invariance is not guaranteed in general. However, if one carefully designs $\Pi$, it can also be achieved via our framework. Moreover, we also suggest a feasible solution to constructing a rotation-invariant operator $\mathcal{A}$ in Theorem 1. In our construction, $\Pi$ first isotropically pools over the squares of all directional derivatives, and then maps the summarized information through another scalar function $f$. We refer interested readers to [39] for more group invariance in differential forms.

**Universal Approximation.** Convolution, formally known as the linear shift-invariant operator, has served as one of the most prevalent signal processing tools in the vision domain. Given two (real-valued) signals $f$ and $g$, we denote their convolution as $g \star f = f \star g$. In this section, we examine the expressiveness of our INSP-Net (Eq. 3) by showing it can represent any convolutional filter. To draw this conclusion, we first present an informal version of our main results as follows:

**Theorem 2.** *(Informal statement) For every real-valued function $g : \mathbb{R}^m \to \mathbb{R}$, there exists a polynomial $p(x_1, \cdots, x_m)$ with real coefficients such that $p(\nabla) f$ can uniformly approximate $g \star f$ by arbitrary precision for all real-valued signals $f$.*

The formal statement and proof can be found in Appendix B. Theorem 2 involves the notion of polynomials in partial differential operators (see more details in Appendix B). $p(\nabla)f$ in turn can be written as a *linear* combination of high-order derivatives of $f$ (a special case of Eq. 3 when $\Pi$ is linear). The key step to prove Theorem 2 is applying Stone-Weierstrass approximation theorem on the Fourier domain. However, we note that functions obtained by the Fourier transform are generally complex functions. The prominence of our proof is that we can constrain the range of the polynomial coefficients into the real domain, which makes it implementable via a common deep learning infrastructure. The implication of Theorem 2 is that the mapping between convolution and derivative is as simple as a linear transformation. Recent works [41, 42, 43] show the converse argument that derivatives can be approximated via a linear combination of *discrete* convolution. Theorem 2 establishes the equivalence between differential operator and convolution in the *continuous* regime. In our proof, $k$-th order derivatives correspond to $k$-th order monomial in the spectral domain. Fitting convolution using derivatives amounts to approximating spectrum via polynomials. This implies higher degree of polynomial induces closer approximation. Since $p(\nabla)$ is not difficult to be approximated by a neural network $\Pi_{\boldsymbol{\theta}}$, we can easily derive the next result Corollary 3.

**Corollary 3.** *For every real-valued function $g$, there exists a neural network $\Pi_{\boldsymbol{\theta}}$ such that $\Psi = \mathcal{A}\Phi$ (Eq. 3) can uniformly approximate $g \star \Phi$ by arbitrary precision for every real-valued signals $\Phi$.*

As we discussed in Theorem 1, $\mathcal{A}$ are constantly shift-invariant. This means when approximating a convolutional kernel, the trajectory of $\mathcal{A}$ is restricted into the shift-invariant space. Moreover, we emphasize that INSP-Net is far more expressive than convolutional kernels since $\Pi_{\boldsymbol{\theta}}$ can also fit any nonlinear continuous functions due to the universal approximation theorem [44, 45, 46].

### 3.3 Building CNNs for Implicit Neural Representations

Convolutional Neural Networks (CNN) are capable of extracting informative semantics by only piling up basic signal processing operators. This motivates us to build CNNs based on INSP-Net that can directly run on INRs for high-level downstream tasks. In fact, to simulate *exact* convolution, our Theorem 2 suggests simplify $\Pi_{\boldsymbol{\theta}}$ to a linear mapping. Then our former computational paradigm Eq. 3 is changed to:

$$\Psi(\boldsymbol{x}) := p(\nabla)\Phi(\boldsymbol{x}) = \theta_0\Phi(\boldsymbol{x}) + \boldsymbol{\theta}_1^\top\nabla\Phi(\boldsymbol{x}) + \boldsymbol{\theta}_2^\top\nabla^2\Phi(\boldsymbol{x}) + \cdots + \boldsymbol{\theta}_k^\top\nabla^k\Phi(\boldsymbol{x}) + \cdots, \quad (5)$$

where $\boldsymbol{\theta}_k \in \mathbb{R}^{\binom{k+m-1}{k}}$ are parameters of the operator $p(\nabla)$. We name this special case of Eq. 3 as INSP-Conv. One plausible implementation of INSP-Conv is to employ a one-layer MLP to represent $\Pi_{\boldsymbol{\theta}}$. When $\mathcal{A} = p(\nabla)$, INSP-Conv preserves both linearity and shift-invariance when evolving during the training. We propose to repeatedly apply INSP-Conv with non-linearity to INRs that mimics a CNN-like architecture. We name this class of CNNs composed by multi-layer INSP-Conv (Eq. 5) as INSP-ConvNet. Previous works [47, 48] extracting semantic features from INR either lack local information by point-wisely mapping INR's intermediate representation to a semantic space or explicitly rasterize INR into regular grids. To the best of our knowledge, it is the first time that one can run a CNN directly on an implicit representation thanks to closed-formness of INSP-Net. The overall architecture of INSP-ConvNet can be formulated as:

$$\text{ConvNet}[\Phi](\boldsymbol{x}) = \mathcal{A}^{(L)} \cdot \sigma \circ \mathcal{A}^{(L-1)} \cdot \sigma \circ \cdots \circ \mathcal{A}^{(2)} \cdot \sigma \circ \mathcal{A}^{(1)} \cdot \Phi(\boldsymbol{x}), \quad (6)$$

where $\sigma$ is an element-wise non-linear activation, $L$ is the number of INSP-Net layers, and $\Phi$ is an input INR. We use the symbol $\cdot$ to denote operator functioning, and $\circ$ to denote function composition. Due to page limit, we defer detailed introduction to INSP-ConvNet to Appendix C.

## 4 Related Work

### 4.1 Implicit Neural Representation

Implicit Neural Representation (INR) represents signals by continuous functions parameterized by multi-layer perceptrons (MLPs) [28, 27], which is different from traditional discrete representations (e.g., pixel, mesh). Compared with other representations, the continuous implicit representations are capable of representing signals at infinite resolution and have become prevailing to be applied upon image fitting [28], image compression [1, 49] and video compressing [3]. In addition, INR has been applied to more efficient and effective shape representation [4, 5, 6, 7, 8, 9, 10, 11], texture mapping [50, 51], inverse problems [12, 2, 13, 14] and generative models [15, 16, 17, 18, 19, 20, 21, 22]. There are also efforts speeding up the fitting of INRs [52] and improving the representation efficiency [53]. Nowadays, editing and manipulating multi-media objects gains increasing interest and demand [54]. Thus, signal processing on implicit neural representation is essentially an important task worth investigating.

### 4.2 Editable Implicit Fields

Editing implicit fields has recently attracted much research interest. Several methods have been proposed to allow editing the reconstructed 3D scenes by rearranging the objects or manipulating the shape and appearance. One line of work alters the structure and color of objects by conditioning latent codes for different characteristics of the scene [25, 20, 21, 26]. Another direction involves discretizing the continuous fields. By converting the implicit fields into pixels or voxels, traditional image and voxel editing techniques [55, 56] can be applied effortlessly. These approaches, however, are not capable of directly performing signal processing on continuous INRs. Functa [54] can use a latent code to control implicit funtions. NID [57] represents neural fields as a linear combination of implicit functional basis, which enables editing by change of sparse coefficients. However, such editing scheme suffers from limited flexibility. Recently, NFGP [58] proposes to use neural fields for geometry processing by exploring various geometric regularization. INS [59] distills stylized features into INRs via the neural style transfer framework [60]. Our INSP-Net that makes smart use of closed-form differential operators does not require neither additional per-scene fine-tuning nor discretization to grids.

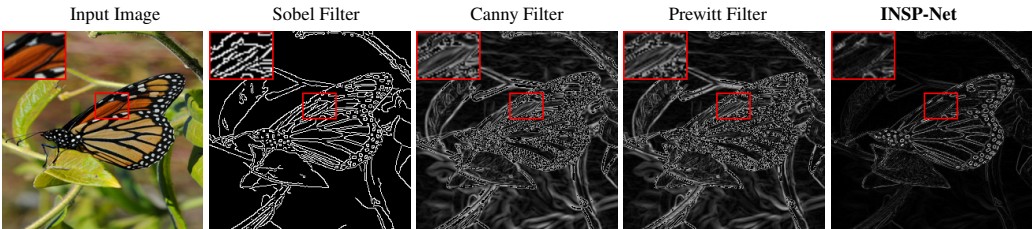

Figure 3: Edge detection. We fit the natural images with SIREN and use our INSP-Net to process implicitly into a new INR that can be decoded into edge maps.

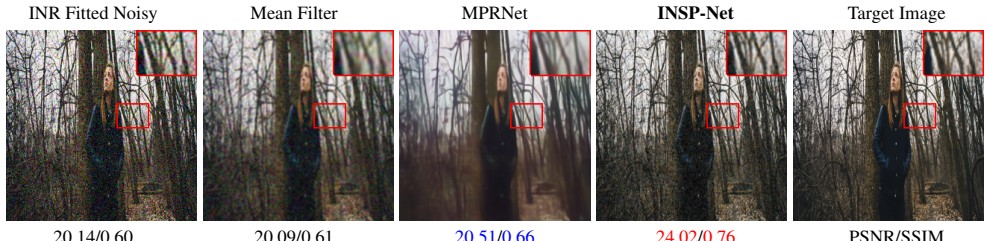

Figure 4: Image denoising. We fit the noisy images with SIREN and train our INSP-Net to process implicitly into a new INR that can be decoded into natural clear images.

## 4.3 PDE based Image Processing

Partial differential equations (PDEs) have been successfully applied to many tasks in image processing and computer vision, such as image enhancement [61, 62, 63], segmentation [64, 40], image registration [65], saliency detection [66] and optical flow computation [67]. Early traditional PDEs are written directly based on mathematical and physical understanding of the PDEs (e.g., anisotropic diffusion [61], shock filter [62] and curve evolution based equations [68, 69, 70]). Variational design methods [63, 71, 70] start from an energy function describing the desired properties of output images and compute the Euler-Lagrange equation to derive the evolution equations. Learning-based attempts [40, 66] build PDEs from image pairs based on the assumption (without proof) that PDEs could be written as linear combinations of fundamental differential invariants. Although it might be feasible to let INRs solve this bunch of signal processing PDEs, one needs to per-case re-fit an INR with an additional temporal axis, which is presumably sampling inefficient. The multi-layer structure appearing in INSP-Net can be viewed as an unfolding network [72, 73] of the Euler method to solve time-variant PDEs [74]. We elaborate this connection in Appendix D.

## 5 Experiments

In this section, we evaluate the proposed **INSP** framework on several challenging tasks, using different combinations of $\Pi$. First, we build low-level image processing filters using either hand-crafted or learnable $\Pi$. Then, we construct convolutional neural networks with our INSP-ConvNet framework and validate its performance on image classification. More results and implementation details are provided in the Appendix E, F.

### 5.1 Low-Level Vision for Implicit Neural Images

For low-level image processing, we operate on natural images from Set5 dataset [75], Set14 dataset [76], and DIV-2k dataset [77]. Originally designed for super-resolution, the images are diverse in style and content. Note that the unprocessed images presented in figures are the images decoded from unprocessed INRs.

Since our method operates directly on INRs, we firstly fit the images with INRs and then feed the INRs into our framework. The final output is another INR which can be decoded into desired images. The training set of our method consists of 90 examples of INRs, where each INR is built on SIREN [28] architectures.

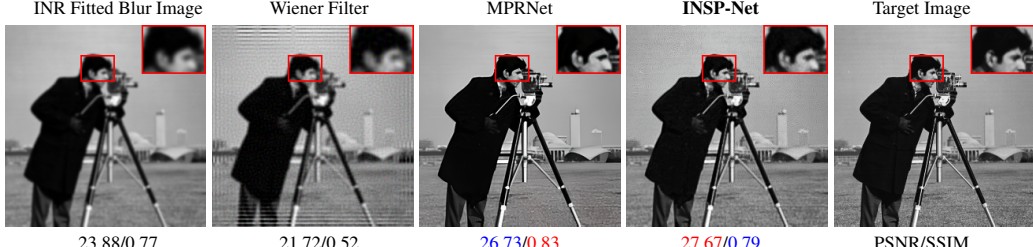

| INR Fitted Blur Image | Wiener Filter | MPRNet | **INSP-Net** | Target Image |
|---|---|---|---|---|
| 23.88/0.77 | 21.72/0.52 | 26.73/0.83 | 27.67/0.79 | PSNR/SSIM |

Figure 5: Image deblurring. We fit the blurred images with SIREN and train our INSP-Net to process implicitly into a new INR that can be decoded into clear natural images.

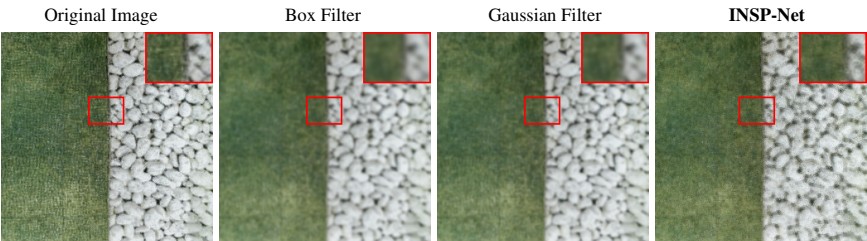

Figure 6: Image blurring. We fit the natural images with SIREN and train our INSP-Net to process implicitly into a new INR that can be decoded into blurred images.

**Edge Detection**    Since the edges correspond to gradients in the images, using gradients of INRs to obtain edges is straightforward. $\theta_1$ is set to 1 while other coefficients are set to 0. We provide visual comparisons against Sobel filter [78], Canny detector [79] and Prewitt operator [80] in Fig. 3.

**Image Denoising**    For classical image denoising filters, we compare against the median filter and mean filter. We use MPRNet [81] as a learning-based baseline. The input noisy images are synthesized using additive gaussian noise. Visual results are provided in Fig. 4.

**Image Blurring**    Image blurring is a low-pass filtering operation. We provide a visual comparison against classical filters including $3 \times 3$ box filter and $3 \times 3$ gaussian filter. The target images used for training our INSP-Net are the results of the Gaussian filter. Visual results are provided in Fig. 6.

**Image Deblurring**    We compare the proposed method with both traditional algorithms (e.g., wiener filter [82]) and learning-based algorithms(e.g., MPRNet [81]). We synthesize blurry images using Gaussian filters. As shown in Fig. 5, Wiener Filter produce severe artifacts and MPRNet successfully reconstructs clear textures. INSP-Net is capable of generating competitive results against MPRNet and outperforms the Wiener Filter.

**Image Inpainting**    We conduct two kinds of experiments in image inpainting, to inpaint 30% random masked pixels or to remove the texts ("INSP-Net"). Comparison methods include mean filter, median filter, and LaMa [83]. LaMa is a learning-based method using Fourier convolution for inpainting. As shown in Fig. 7, mean filter and median filter partially restore the masked pixels, but severely hurt the visual quality of the rest parts. Also, they can not handle the text region. LaMa successfully removes the text and inpaint the masked pixels. Our proposed method largely outperforms the filter-based algorithms and performs as well as the LaMa.

## 5.2   Geometry Processing on Signed Distance Function

We demonstrate that the proposed **INSP** framework is not only capable of processing images, but also capable of processing geometry. Signed Distance Function (SDF) [25] is adopted to represent geometries in this section. We first fit an SDF from a point cloud following the training loss proposed in [28, 7]. Then we train our INSP-Net to simulate a Gaussian-like filter similar to image blurring. Afterwards, we apply the trained INSP-Net to process the specified INR. When visualization, we use marching cube algorithm to extract meshes from SDF. We choose Thai Statue, Armadillo, and Dragon from Stanford 3D Scanning Repository [84, 85, 86, 87] to demonstrate our results. Fig. 8

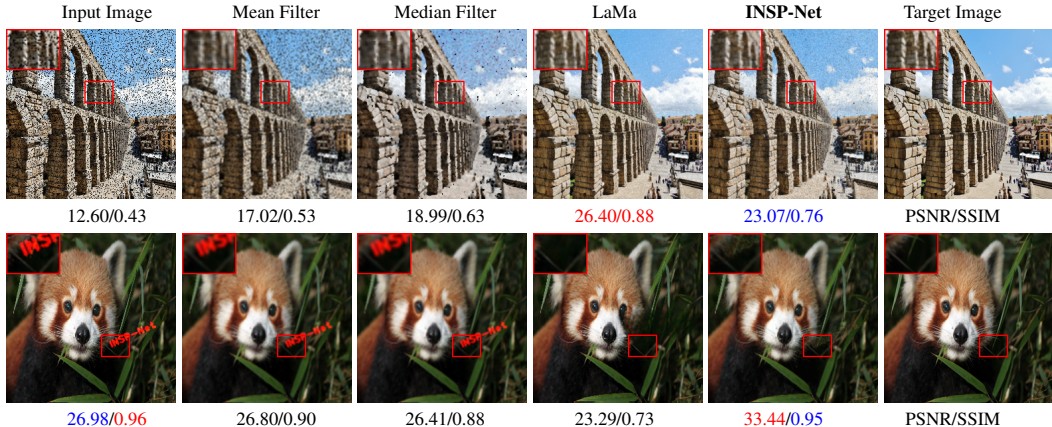

| Input Image | Mean Filter | Median Filter | LaMa | **INSP-Net** | Target Image |
|---|---|---|---|---|---|
| 12.60/0.43 | 17.02/0.53 | 18.99/0.63 | 26.40/0.88 | 23.07/0.76 | PSNR/SSIM |
| 26.98/0.96 | 26.80/0.90 | 26.41/0.88 | 23.29/0.73 | 33.44/0.95 | PSNR/SSIM |

Figure 7: Image inpainting. We fit the input images with SIREN and train our INSP-Net to process implicitly into a new INR that can be decoded into natural images. Note that LaMa requires explicit masks to select the regions for inpainting and the masks are roughly provided.

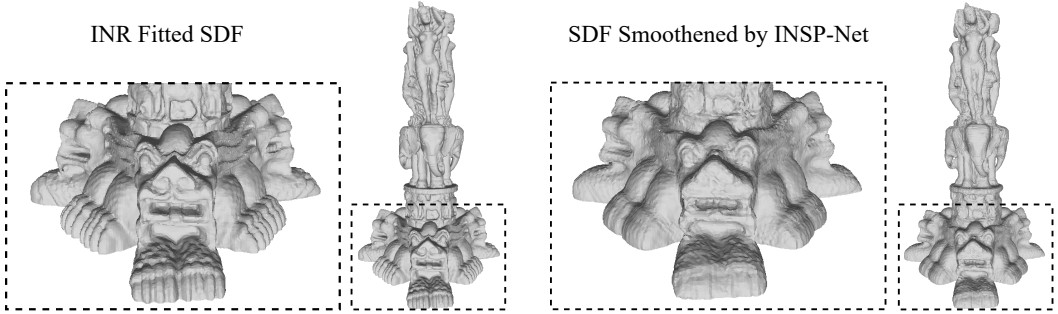

Figure 8: Left: unprocessed geometry decoded from an unprocessed INR. Right: smoothened geometry decoded from the output INR of our INSP-Net. Best view in a zoomable electronic copy.

exhibits our results on Thai Statue. Our method is able to smoothen the surface of the geometry and erase high-frequency details acting as if a low-pass filter. We defer more results to Fig. 14.

## 5.3 Classification on Implicit Neural Representations

We demonstrate that the proposed **INSP** framework is not only capable to express low-level image processing filters, but also supports high-level tasks such as image classification. To achieve this goal, we construct a 2-layer INSP-ConvNet. The INSP-ConvNet consists of 2 INSP-Net layers. Each of them decomposes the INR via the differential operator and combines them with learnable $\Pi$. We build another 2-layer depthwise ConvNets running on pixels as the baseline for a fair comparison, since it has comparable expressiveness to our INSP-ConvNet in theory. We also build a PCA + SVM method and an MLP classifier that directly classify INRs according to (vectorized) weight matrices.

We evaluate the proposed INSP-ConvNet on MNIST ($28 \times 28$ resolution) and CIFAR-10 ($32 \times 32$ resolution) datasets, respectively. For each dataset, we will firstly fit each image into an implicit representation using SIREN [28]. Both experiments take 1000 epochs to optimize with AdamW optimizer [88] and a learning rate of $10^{-4}$. Results are shown in Tab. 1.

| Accuracy | Depthwise CNN | PCA + SVM | MLP classifier | INSP-ConvNet |
|---|---|---|---|---|
| MNIST | 87.6% | 11.3% | 9.8% | 88.1% |
| CIFAR-10 | 59.5% | 9.4% | 10.1% | 62.5% |

Table 1: Quantitative Results of Image Classification. All methods except "Depthwise CNN" operate on the parameter of INR directly, while "Depthwise CNN" operates on images decoded from INR.

We categorize Depthwise CNN as explicit method, which requires to extract the image grids from INRs before classification. PCA + SVM and MLP classifier working on the network parameter space can be regarded as two straightforward implicit baselines. We find that traditional classifiers can hardly classify INR on weight space due to its high-dimensional unstructured data distribution. Our method, however, can effectively leverage the information implicitly encoded in INRs by exploiting their derivatives. As a consequence, INSP-ConvNet can achieve classification accuracy on-par with CNN-based explicit method, which validates the learning representation power of INSP-ConvNet.

## 6   Conclusion

**Contribution.**   We present INSP-Net framework, an implicit neural signal processing network that is capable of directly modifying an INR without explicit decoding. By incorporating differential operators on INR, we can instantiate the INR signal operator as a composition of computational graphs approximating any continuous convolution filter. Furthermore, we make the first effort to build a convolutional neural network that implicitly runs on INRs. While all other methods run on discrete grids, our experiment demonstrates our INSP-Net can achieve competitive results with entirely implicit operations.

**Limitations.**   (Theory) Our theory only guarantees the expressiveness of convolution by allowing infinite sequence approximation. Construction of more expressive operators and more effective parameterization for convolution remain widely open questions. (Practice) INSP-Net requires the computation of high-order derivatives which is neither memory efficient nor numerically stable. This hinders the scalability of our INSP-ConvNet that requires recursive computation of derivatives. Addressing how to reconstruct INRs in a scalable manner is beyond the scope of this paper. All INRs used in our experiments are fitted by per-scene optimization.

## Acknowledgement

Z. Wang is in part supported by an NSF Scale-MoDL grant (award number: 2133861).

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
