# A Proof of Theorem 1

To begin with, we give the formal definitions of translation and rotation group, along with the notion of shift invariance and rotation invariance.

**Definition A.1.** *(Translation Group & Shift Invariance) Translation group $\mathbb{T}(m)$ is a transformation group isomorphic to $m$-dimension Euclidean space, where each group element $T_{\boldsymbol{v}}$ transforms a vector $\boldsymbol{x} \in \mathbb{R}^m$ by $T_{\boldsymbol{v}}(\boldsymbol{x}) = \boldsymbol{x} + \boldsymbol{v}$. An operator $\mathcal{A}$ is said to be shift-invariant if $\mathcal{A}(\Phi \circ T_{\boldsymbol{v}})(\boldsymbol{x}) = \mathcal{A}\Phi(T_{\boldsymbol{v}}(\boldsymbol{x})) = \mathcal{A}\Phi(\boldsymbol{x} + \boldsymbol{v})$.*

**Definition A.2.** *(Rotation Group & Rotation Invariance) Rotation group $\mathbb{SO}(m)$ is a transformation group also known as the special orthogonal group, where each group element $\boldsymbol{R} \in \mathbb{R}^{m \times m}$ satisfying $\boldsymbol{R}^\top \boldsymbol{R} = \boldsymbol{I}$ transforms a vector $\boldsymbol{x} \in \mathbb{R}^m$ by $\boldsymbol{R}(\boldsymbol{x})$. An operator $\mathcal{A}$ is said to be rotation-invariant if $\mathcal{A}(\Phi \circ \boldsymbol{R})(\boldsymbol{x}) = \mathcal{A}\Phi(\boldsymbol{R}\boldsymbol{x})$.*

*Proof.* (Shift Invariance) To show the shift invariance of our model Eq. 3, it is equivalent to show any differential operators are shift-invariant. For the first-order derivatives (gradients), we consider arbitrary shift operator $T_{\boldsymbol{v}} \in \mathbb{T}$, by chain rule we will have:

$$\nabla[\Phi \circ T_{\boldsymbol{v}}](\boldsymbol{x}) = \left[\frac{d(\boldsymbol{x} + \boldsymbol{v})}{d\boldsymbol{x}}\right]^\top \nabla\Phi(\boldsymbol{x} + \boldsymbol{v}) = \nabla\Phi(\boldsymbol{x} + \boldsymbol{v}), \tag{7}$$

where the Jacobian matrix of $T_{\boldsymbol{v}}(\boldsymbol{x})$ is an identity matrix. Eq. 7 implies that gradient operator is shift-invariant. By induction, any high-order differential operators must also be shift-invariant:

$$\nabla^k[\Phi \circ T_{\boldsymbol{v}}](\boldsymbol{x}) = \nabla^k\Phi(\boldsymbol{x} + \boldsymbol{v}), \tag{8}$$

Therefore, we can conclude $\Pi(\Phi, \nabla\Phi, \nabla^2\Phi, \cdots)$ is shift-invariant for any $\Pi$ combining derivatives in any form.

(Rotation Invariance) By Lemma A.1, given arbitrary function $\Phi : \mathbb{R}^m \to \mathbb{R}$, and for every rotation matrix $\boldsymbol{R} \in \mathbb{SO}(m)$, we can compute the $k$-th derivatives as:

$$\mathrm{vec}\left(\nabla^k[\Phi \circ \boldsymbol{R}](\boldsymbol{x})\right) = \boldsymbol{R}^{\top \otimes k} \mathrm{vec}\left(\nabla\Phi(\boldsymbol{R}\boldsymbol{x})\right). \tag{9}$$

Then adopting properties of Kronecker product [89], the norm of $\nabla^k[\Phi \circ \boldsymbol{R}](\boldsymbol{x})$ can be written as:

$$\|\nabla^k[\Phi \circ \boldsymbol{R}](\boldsymbol{x})\|_F^2 = \mathrm{Tr}\left[\mathrm{vec}\left(\nabla\Phi(\boldsymbol{R}\boldsymbol{x})\right)^\top \boldsymbol{R}^{\otimes k} \boldsymbol{R}^{\top \otimes k} \mathrm{vec}\left(\nabla\Phi(\boldsymbol{R}\boldsymbol{x})\right)\right] \tag{10}$$

$$= \mathrm{vec}\left(\nabla\Phi(\boldsymbol{R}\boldsymbol{x})\right)^\top \left(\boldsymbol{R}^{\otimes k-1} \otimes \boldsymbol{R}\right)\left(\boldsymbol{R}^{\top \otimes k-1} \otimes \boldsymbol{R}^\top\right) \mathrm{vec}\left(\nabla\Phi(\boldsymbol{R}\boldsymbol{x})\right) \tag{11}$$

$$= \mathrm{vec}\left(\nabla\Phi(\boldsymbol{R}\boldsymbol{x})\right)^\top \left(\left(\boldsymbol{R}^{\otimes k-1} \boldsymbol{R}^{\top \otimes k-1}\right) \otimes \boldsymbol{I}\right) \mathrm{vec}\left(\nabla\Phi(\boldsymbol{R}\boldsymbol{x})\right) \tag{12}$$

$$= \cdots = \mathrm{vec}\left(\nabla\Phi(\boldsymbol{R}\boldsymbol{x})\right)^\top \boldsymbol{I}^{\otimes k} \mathrm{vec}\left(\nabla\Phi(\boldsymbol{R}\boldsymbol{x})\right) = \|\nabla\Phi(\boldsymbol{R}\boldsymbol{x})\|_F^2 \tag{13}$$

where Eq. 11 is due to the fact $(\boldsymbol{A} \otimes \boldsymbol{B})^\top = \boldsymbol{A}^\top \otimes \boldsymbol{B}^\top$, Eq. 12 is because of $(\boldsymbol{A} \otimes \boldsymbol{B})(\boldsymbol{C} \otimes \boldsymbol{D}) = \boldsymbol{A}\boldsymbol{C} \otimes \boldsymbol{B}\boldsymbol{D}$, and Eq. 13 is yielded by applying the orthogonality of $\boldsymbol{R}$ and repeating step Eq. 12 to Eq. 13. Therefore, for every integer $k > 0$, operator $\|\nabla^k\Phi(\boldsymbol{x})\|_2^2$ is rotation-invariant. Hence, $\Pi = f\left(\left\|\begin{bmatrix}\Phi(\boldsymbol{x}) & \nabla\Phi(\boldsymbol{x}) & \nabla^2\Phi(\boldsymbol{x}) & \cdots\end{bmatrix}\right\|_F\right) = f\left(\sqrt{\sum_{k=0}\|\nabla^k\Phi(\boldsymbol{x})\|_2^2}\right)$ is also rotation-invariant. $\square$

Below we supplement the Lemma A.1 used to prove Theorem 1.

**Lemma A.1.** *Suppose given function $f : \mathbb{R}^m \to \mathbb{R}$ and arbitrary linear transformation $\boldsymbol{A} \in \mathbb{R}^{m \times m}$, then $\mathrm{vec}\left(\nabla^k[f \circ \boldsymbol{A}](\boldsymbol{x})\right) = \boldsymbol{A}^{\top \otimes k} \mathrm{vec}\left(\nabla^k f(\boldsymbol{A}\boldsymbol{x})\right)$ for $\forall k \geq 0$.*

*Proof.* $\mathrm{vec}\left(\nabla^k[f \circ \boldsymbol{A}](\boldsymbol{x})\right) = \boldsymbol{A}^{\top \otimes k} \mathrm{vec}\left(\nabla^k f(\boldsymbol{A}\boldsymbol{x})\right)$ trivially holds for $k = 0, 1$. Then we prove Lemma A.1 by induction. Suppose the $(j-1)$-th case satisfies the equality: $\mathrm{vec}\left(\nabla^{j-1}[f \circ \boldsymbol{A}](\boldsymbol{x})\right) = \boldsymbol{A}^{\top \otimes j-1} \mathrm{vec}\left(\nabla^{j-1} f(\boldsymbol{A}\boldsymbol{x})\right)$, then consider the $j$-th case:

$$\nabla^j[f \circ \boldsymbol{A}](\boldsymbol{x}) = \nabla \mathrm{vec}\left(\nabla^{j-1}[f \circ \boldsymbol{A}](\boldsymbol{x})\right) = \nabla \boldsymbol{A}^{\top \otimes j-1} \mathrm{vec}\left(\nabla^{j-1} f(\boldsymbol{A}\boldsymbol{x})\right) = \boldsymbol{A}^\top \nabla^j f(\boldsymbol{A}\boldsymbol{x}) \boldsymbol{A}^{\otimes j-1},$$

where the first equality is done by reshaping the $m^j$ tensor to be an $m \times m^{j-1}$ Jacobian matrix, the second equality is due to the induction hypothesis, and the third equality is an adoption of chain rule. Due to the fact $\mathrm{vec}(\boldsymbol{A}\boldsymbol{B}\boldsymbol{C}) = (\boldsymbol{C}^\top \otimes \boldsymbol{A}) \mathrm{vec}(\boldsymbol{B})$, we have $\mathrm{vec}\left(\nabla^j[f \circ \boldsymbol{A}](\boldsymbol{x})\right) = \boldsymbol{A}^{\top \otimes j} \mathrm{vec}\left(\nabla^j f(\boldsymbol{A}\boldsymbol{x})\right)$. Then by induction, we can conclude the proof. $\square$

# B    Proof of Theorem 2

For a sake of clarity, we first introduce few notations in algebra and real analysis. We use $C^k(\mathcal{X}, \mathbb{R})$ to denote the $k$-th differentiable functions defined over domain $\mathcal{X}$, $W^{k,p}(\mathcal{X}, \mathbb{R})$ to denote the $k$-th differentiable and $L^p$ integrable Sobolev space over domain $\mathcal{X}$. We use notation $\mathcal{A}[f]$ to denote the image of function (say $f$) under the transformation of an operator (say $\mathcal{A}$). We use symbol $\circ$ to denote function composition (e.g., $f \circ g(\boldsymbol{x}) = f(g(\boldsymbol{x}))$). We use dot-product $\cdot$ between two functions (say $f$ and $g$) to represent element-wise multiplication of function values (say $f \cdot g(\boldsymbol{x}) = f(\boldsymbol{x}) \cdot g(\boldsymbol{x})$). Besides, we list the following definitions and assumptions:

**Definition B.1.** *(Polynomial) We use $\mathbb{R}[x_1, \cdots, x_m]$ to represent the multivariate polynomials in terms of $x_1, \cdots, x_m$ with real coefficients. We write a (monic) multivariate monomial $m(x_1, \cdots, x_m) = x_1^{n_1} x_2^{n_2} \cdots x_m^{n_m}$ as $m(\boldsymbol{x}) = \boldsymbol{x^n}$ where $\boldsymbol{n} = \begin{bmatrix} n_1 & \cdots & n_m \end{bmatrix} \in \mathbb{N}^m$. Then we denote a polynomial as $p(\boldsymbol{x}) = a_1 \boldsymbol{x^{n_1}} + \cdots + a_d \boldsymbol{x^{n_d}} \in \mathbb{R}[x_1, \cdots, x_m]$ where $\boldsymbol{x^{n_i}}$ denotes the $i$-th multivariate monomial and $a_i \in \mathbb{R}$ is the corresponding coefficient.*

**Definition B.2.** *(Differential Operator) Suppose a compact set $\mathcal{X} \subseteq \mathbb{R}^m$. we denote $\mathcal{D^n}$ : $C^\infty(\mathcal{X}, \mathbb{R}) \to C^\infty(\mathcal{X}, \mathbb{R})$ as the high-order differential operator associated with indices $\boldsymbol{n} \in \mathbb{N}^m$:*

$$\mathcal{D^n}[f] = \frac{\partial^{\|\boldsymbol{n}\|_1}}{\partial x_1^{n_1} \cdots \partial x_m^{n_m}} f. \tag{14}$$

**Definition B.3.** *We define polynomial in gradient operator as: $p(\nabla) = p\left(\frac{\partial}{\partial x_1}, \cdots, \frac{\partial}{\partial x_m}\right) = a_1 \mathcal{D^{n_1}} + \cdots + a_d \mathcal{D^{n_d}} \in \mathbb{R}[x_1, \cdots, x_m]$ where $\mathcal{D^{n_i}}$ denotes the $\boldsymbol{n}_i$-th order partial derivative (Definition B.2) and $a_i \in \mathbb{R}$ is the corresponding coefficient.*

**Remark B.1.** *The mapping between $p(\boldsymbol{x})$ and $p(\nabla)$ is a ring homomorphism from polynomial ring $\mathbb{R}[x_1, \cdots, x_m]$ to the ring of endomorphism defined over $C^\infty(\mathcal{X}, \mathbb{R})$.*

**Definition B.4.** *(Fourier Transform) Given real-valued function $f : \mathbb{R}^m \to \mathbb{C}$ that satisfies Dirichlet condition[5], then Fourier transform $\mathcal{F}$ is defined as:*

$$\mathcal{F}[f](\boldsymbol{w}) = \int_{\mathbb{R}^m} f(\boldsymbol{x}) \exp(-2\pi i \boldsymbol{w}^\top \boldsymbol{x}) d\boldsymbol{x}. \tag{15}$$

*Inverse Fourier transform $\mathcal{F}^{-1}$ exists and has the form of:*

$$f(\boldsymbol{x}) = \int_{\mathbb{R}^m} \mathcal{F}[f](\boldsymbol{w}) \exp(2\pi i \boldsymbol{w}^\top \boldsymbol{x}) d\boldsymbol{w}. \tag{16}$$

**Definition B.5.** *(Convolution) Given two real-valued functions $f : \mathbb{R}^m \to \mathbb{R}$ and $g : \mathbb{R}^m \to \mathbb{R}$, convolution between $f$ and $g$ is defined as:*

$$(f \star g)(\boldsymbol{x}) = \int_{\mathbb{R}^m} f(\boldsymbol{x} - \boldsymbol{\xi}) g(\boldsymbol{\xi}) d\boldsymbol{\xi}. \tag{17}$$

*Then we denote $f \star g = \mathcal{T}_g[f]$, where $\mathcal{T}_g$ represents a convolutional operator associated with the function $g$.*

We make the following mild assumptions on the signals and convolutional operators, which are widely satisfied by the common signals and systems.

**Assumption B.6.** *(Band-limited Signal Space) Define the signal space $\mathcal{S}$ as a Sobolev space $W^{\infty,1}(\mathbb{R}^m, \mathbb{R})$ of real-valued functions such that for $\forall f \in \mathcal{S}$:*

*(I)  $f \in C^\infty(\mathbb{R}^m, \mathbb{R})$ is continuous and smooth over $\mathbb{R}^m$.*

*(II)  $f$ satisfies the Dirichlet condition.*

*(III)  $f$ has a limited width of spectrum: there exists a compact subset $\mathcal{W} \subset \mathbb{R}^m$ such that $|\mathcal{F}[f](\boldsymbol{w})| = 0$ if $\boldsymbol{w} \notin \mathcal{W}$, and $\int_{\mathcal{W}} |\mathcal{F}[f](\boldsymbol{w})| d\boldsymbol{w} < \infty$.*

---

[5]Dirichlet condition guarantees Fourier transform exists: (1) The function is $L_1$ integrable over the entire domain. (2) The function has at most a countably infinite number of infinte minima or maxma or discontinuities over the entire domain.

**Assumption B.7.** *(Convolution Space) Define a convolutional operator space $\mathcal{T}$ such that $\forall \mathcal{T}_g \in \mathcal{T}$:*

   *(IV) $g : \mathbb{R}^m \to \mathbb{R}$ is real-valued function.*

   *(V) $\mathcal{F}[g] \in C(\mathbb{R}^m, \mathbb{R})$ has a continuous spectrum.*

Before we prove Theorem 2, we enumerate the following results as our key mathematical tools:

First of all, we note the following well-known result without a proof.

**Lemma B.1.** *(Convolution Theorem) For every $\mathcal{T}_g \in \mathcal{T}$, it always holds that $\mathcal{F} \circ \mathcal{T}_g[f](\boldsymbol{w}) = \mathcal{F}[f](\boldsymbol{w}) \cdot \mathcal{F}[g](\boldsymbol{w})$.*

Next, we present Stone-Weierstrass Theorem as our Lemma B.2 as below.

**Lemma B.2.** *(Stone-Weierstrass Theorem) Suppose $\mathcal{X}$ is a compact metric space. If $\mathcal{A} \subset C(\mathcal{X}, \mathbb{R})$ is a unital sub-algebra which separates points in $\mathcal{X}$. Then $\mathcal{A}$ is dense in $C(\mathcal{X}, \mathbb{R})$.*

A straightforward corollary of Lemma B.2 is the following Lemma B.3.

**Lemma B.3.** *Let $\mathcal{X} \subset \mathbb{R}^m$ be a compact subset of $\mathbb{R}^m$. For every $\epsilon > 0$, there exists a polynomial $p(\boldsymbol{x}) \in \mathbb{R}[x_1, \cdots, x_m]$ such that $\sup_{\boldsymbol{x} \in \mathcal{X}} |f(\boldsymbol{x}) - p(\boldsymbol{x})| < \epsilon$.*

*Proof.* Proved by checking polynomials $\mathbb{R}[x_1, \cdots, x_m]$ form a unital sub-algebra separating points in $\mathcal{X}$, and equipping $C(\mathcal{X}, \mathbb{R})$ with the distance metric $d(f, h) = \sup_{\boldsymbol{x} \in \mathcal{X}} |f(\boldsymbol{x}) - g(\boldsymbol{x})|$. $\qquad \square$

We also provide the following Lemma B.4 to reveal the spectrum-domain symmetry for real-valued signals.

**Lemma B.4.** *Suppose $f$ is a continuous real-valued function satisfying Dirichlet condition. Then $\mathcal{F}[f](\boldsymbol{w}) = \mathcal{F}[f](-\boldsymbol{w})^*$, i.e., the spectrum of real-valued function is conjugate symmetric.*

*Proof.* By the definition of Fourier transform (Definition B.4):

$$\mathcal{F}[f](-\boldsymbol{w}) = \int_{\mathbb{R}^m} f(\boldsymbol{x}) \exp(2\pi i \boldsymbol{w}^\top \boldsymbol{x}) d\boldsymbol{x} = \int_{\mathbb{R}^m} f(\boldsymbol{x})^* \exp(-2\pi i \boldsymbol{w}^\top \boldsymbol{x})^* d\boldsymbol{x} \qquad (18)$$

$$= \left[ \int_{\mathbb{R}^m} f(\boldsymbol{x}) \exp(-2\pi i \boldsymbol{w}^\top \boldsymbol{x}) d\boldsymbol{x} \right]^* = \mathcal{F}[f](\boldsymbol{w})^*, \qquad (19)$$

where Eq. 18 holds because $f$ is a real-valued function. $\qquad \square$

We present Lemma B.5 as below, which reflects the effect of differential operators on the spectral domain.

**Lemma B.5.** *Suppose $f \in C^\infty(\mathbb{R}^m, \mathbb{R})$ is a smooth real-valued function satisfying Dirichlet condition. Then $\mathcal{F} \circ \mathcal{D}^{\boldsymbol{n}}[f](\boldsymbol{w}) = (2\pi i)^{\|\boldsymbol{n}\|_1} \boldsymbol{w}^{\boldsymbol{n}} \cdot \mathcal{F}[f](\boldsymbol{w})$ for every $\boldsymbol{n} \in \mathbb{N}^m$.*

*Proof.* We first show the case of first-order partial derivative. Suppose $h \in C(\mathbb{R}^m, \mathbb{R})$ is $L^1$ integrable (then Fourier transform exists).

$$\frac{\partial}{\partial x_i} h(\boldsymbol{x}) = \frac{\partial}{\partial x_i} \int_{\mathbb{R}^m} \mathcal{F}[h](\boldsymbol{w}) \exp(2\pi i \boldsymbol{w}^\top \boldsymbol{x}) d\boldsymbol{w} \qquad (20)$$

$$= \int_{\mathbb{R}^m} \mathcal{F}[h](\boldsymbol{w}) \frac{\partial}{\partial x_i} \exp(2\pi i \boldsymbol{w}^\top \boldsymbol{x}) d\boldsymbol{w} \qquad (21)$$

$$= 2\pi i \int_{\mathbb{R}^m} w_i \mathcal{F}[h](\boldsymbol{w}) \exp(2\pi i \boldsymbol{w}^\top \boldsymbol{x}) d\boldsymbol{w}. \qquad (22)$$

Then we apply the Fourier transform to Eq. 20, we can obtain:

$$\mathcal{F} \circ \frac{\partial}{\partial x_i}[h](\boldsymbol{w}) = 2\pi i w_i \mathcal{F}[h](\boldsymbol{w}). \qquad (23)$$

Note that $f \in W^{\infty, 1}(\mathbb{R}^m, \mathbb{R})$ ensures all its partial derivatives are differentiable and absolutely integrable. We can recursively apply $\frac{\partial}{\partial x_i}$ to $f$ for $n_i$ times for each $i \in [m]$, and use Eq. 23 above to conclude the proof. $\qquad \square$

Below is the formal statement of our Theorem 2 and its detailed proof.

**Theorem B.6.** *For every $\mathcal{T}_g \in \mathcal{T}$ and arbitrarily small $\epsilon > 0$, there exists a polynomial $p(\boldsymbol{x}) \in \mathbb{R}[x_1, \cdots, x_m]$ such that $\sup_{\boldsymbol{x} \in \mathbb{R}^m} |\mathcal{T}_g[f](\boldsymbol{x}) - p(\nabla)[f](\boldsymbol{x})| < \epsilon$ for all $f \in \mathcal{S}$.*

*Proof.* For every $f \in \mathcal{S}$ and $\mathcal{T}_g \in \mathcal{T}$, by Lemma B.1, one can rewrite:

$$\mathcal{F} \circ \mathcal{T}_g[f](\boldsymbol{w}) = \mathcal{F}[f](\boldsymbol{w}) \cdot \mathcal{F}[g](\boldsymbol{w}) := \hat{f}(\boldsymbol{w})\hat{g}(\boldsymbol{w}), \tag{24}$$

where we use $\hat{f} : \mathbb{R}^m \to \mathbb{C}$ and $\hat{g} : \mathbb{R}^m \to \mathbb{C}$ to denote the Fourier transform of $f$ and $g$, respectively. We can construct an invertible mapping $\phi$ by letting:

$$\phi[\hat{f}](\boldsymbol{w}) = \Re\{\hat{f}(\boldsymbol{w})\} - \Im\{\hat{f}(\boldsymbol{w})\}, \tag{25}$$

$$\phi^{-1}[\tilde{f}](\boldsymbol{w}) = \frac{\tilde{f}(\boldsymbol{w}) + \tilde{f}(-\boldsymbol{w})}{2} - i\frac{\tilde{f}(\boldsymbol{w}) - \tilde{f}(-\boldsymbol{w})}{2}, \tag{26}$$

which is also known as the Hartley transform. By Lemma B.4 (with Assumption (I)(IV)), $\tilde{f} := \phi[\hat{f}]$ and $\tilde{g} := \phi[\hat{g}]$ are both real-valued functions.

Since $\hat{f}$ is only supported in $\mathcal{W}$ (by Assumption (III)), we only consider $\tilde{g}$ within the compact subset $\mathcal{W}$. By Lemma B.3 (with Assumption (V)), there exists a polynomial $\tilde{p}(\boldsymbol{w}) \in \mathbb{R}[w_1, \cdots, w_m] = \tilde{a}_0 + \tilde{a}_1 \boldsymbol{w}_{\boldsymbol{n}_1} + \cdots + \tilde{a}_d \boldsymbol{w}^{\boldsymbol{n}_d}$ such that $\sup_{\boldsymbol{w} \in \mathcal{W}} |\tilde{g}(\boldsymbol{w}) - \tilde{p}(\boldsymbol{w})| < \epsilon/2C$ for every $\epsilon > 0$, where $d$ is the number of monomials in $\tilde{p}$, $\tilde{a}_0, \tilde{a}_1, \cdots, \tilde{a}_d \in \mathbb{R}$ are corresponding coefficients, and $C > 0$ is some constant.

Applying $\phi^{-1}$ to $\tilde{p}$, we will obtain a new (complex-valued) polynomial $\hat{p} := \phi^{-1}[\tilde{p}] \in \mathbb{C}[w_1, \cdots, w_m]$ such that:

$$\Re\{\hat{p}(\boldsymbol{w})\} = \frac{\tilde{p}(\boldsymbol{w}) + \tilde{p}(-\boldsymbol{w})}{2}, \qquad \Im\{\hat{p}(\boldsymbol{w})\} = \frac{\tilde{p}(\boldsymbol{w}) - \tilde{p}(-\boldsymbol{w})}{2}. \tag{27}$$

We observe that the coefficients of $\hat{p}$ satisfy: $\hat{a}_k = \tilde{a}_k$ if $\|\boldsymbol{n}_k\|_1$ is even and $\hat{a}_k = i\tilde{a}_k$ if $\|\boldsymbol{n}_k\|_1$ is odd. Then we bound the difference between $\hat{g}$ and $\hat{p}$ for every $\boldsymbol{w} \in \mathcal{W}$:

$$|\hat{g}(\boldsymbol{w}) - \hat{p}(\boldsymbol{w})| = \left| \left( \frac{\tilde{f}(\boldsymbol{w}) + \tilde{f}(-\boldsymbol{w})}{2} - \frac{\tilde{p}(\boldsymbol{w}) + \tilde{p}(-\boldsymbol{w})}{2} \right) \right. \tag{28}$$

$$\left. - i \left( \frac{\tilde{f}(\boldsymbol{w}) - \tilde{f}(-\boldsymbol{w})}{2} - \frac{\tilde{p}(\boldsymbol{w}) - \tilde{p}(-\boldsymbol{w})}{2} \right) \right| \tag{29}$$

$$\leq \frac{1}{2} \left( \left| \tilde{f}(\boldsymbol{w}) - \tilde{p}(\boldsymbol{w}) \right| + \left| \tilde{f}(-\boldsymbol{w}) - \tilde{p}(-\boldsymbol{w}) \right| \right) \tag{30}$$

$$+ \frac{1}{2} \left( \left| \tilde{p}(\boldsymbol{w}) - \tilde{f}(\boldsymbol{w}) \right| + \left| \tilde{p}(-\boldsymbol{w}) - \tilde{f}(-\boldsymbol{w}) \right| \right) \tag{31}$$

$$\leq \frac{\epsilon}{C}. \tag{32}$$

In the meanwhile, by Lemma B.5 (with Assumption (I), (II)), $\mathcal{F} \circ \mathcal{D}^{\boldsymbol{n}}[f](\boldsymbol{w}) = (2\pi i)^{\|\boldsymbol{n}\|_1} \boldsymbol{w}^{\boldsymbol{n}} \cdot \mathcal{F}[f](\boldsymbol{w})$ for every $\boldsymbol{n} \in \mathbb{N}^m$. Define a sequence $q_{\boldsymbol{n}}(\boldsymbol{w}) = (2\pi i)^{\|\boldsymbol{n}\|_1} \boldsymbol{w}^{\boldsymbol{n}}$, then partial derivatives of $f$ in terms of $\boldsymbol{n} \in \mathbb{N}^m$ can be written as:

$$\mathcal{F} \circ \mathcal{D}^{\boldsymbol{n}}[f](\boldsymbol{w}) = q_{\boldsymbol{n}}(\boldsymbol{w}) \cdot \mathcal{F}[f](\boldsymbol{w}). \tag{33}$$

Next we decompose polynomial $\hat{p}$ in terms of $q_{\boldsymbol{n}}$. Let $a_k = \hat{a}_k/(2\pi i)^{\|\boldsymbol{n}_k\|_1}$, then $\hat{p}(\boldsymbol{w}) = a_0 + a_1 q_{\boldsymbol{n}_1}(\boldsymbol{w}) + \cdots + a_d q_{\boldsymbol{n}_d}(\boldsymbol{w})$. We note that $\{a_k, \forall k \in [d]\}$ must be real numbers since $\hat{a}_k$ is real/imaginary when $\|\boldsymbol{n}_k\|_1$ is even/odd, which coincides with $(2\pi i)^{\|\boldsymbol{n}_k\|_1}$.

By linearity of inverse Fourier transform and Eq. 33, element-wisely multiplying $\sum_{k=0}^d a_k q_{\boldsymbol{n}_k}$ to $\hat{f}$ will lead to a transform on the spatial domain:

$$\mathcal{F}^{-1}\left[ \sum_{k=0}^d a_k q_{\boldsymbol{n}_k} \cdot \mathcal{F}[f] \right] = \sum_{k=0}^d a_k \mathcal{D}^{\boldsymbol{n}_k} f := p(\nabla)[f], \tag{34}$$

---

**Algorithm 1** Forward pass of INSP-ConvNet

---

1: **Input**: An INR network $\Phi(\boldsymbol{x}) : \mathbb{R}^m \to \mathbb{R}$, convolutional operator weights $\boldsymbol{\theta}^{(l)} \in \mathbb{R}^M$ and an input coordinate $\boldsymbol{x}$.
2: **Output**: Value at $\boldsymbol{x}$ of INR $\text{ConvNet}[\Phi]$ processed by INSP-ConvNet.
3: $y^{(0)} \leftarrow \Phi(\boldsymbol{x})$
4: **for** $l = 1, \cdots, L$ **do**
5: $\quad \hat{y}^{(l)} \leftarrow \left[ y^{(l-1)} \quad \frac{\partial y^{(l-1)}}{\partial \boldsymbol{x}}^\top \quad \frac{\partial^2 y^{(l-1)}}{\partial \boldsymbol{x}^2}^\top \quad \cdots \quad \frac{\partial^K y^{(l-1)}}{\partial \boldsymbol{x}^K}^\top \right] \boldsymbol{\theta}^{(l)}$  $\quad \triangleright$ Convolutional layer
6: $\quad y^{(l)} \leftarrow \text{ReLU}(\text{InstanceNorm1D}(\hat{y}^{(l)}))$  $\quad \triangleright$ Non-linearity and normalization
7: **end for**
8: **return** $y^{(L)}$.

---

where we define polynomial $p(\nabla) := a_0 + a_1 \mathcal{D}^{\boldsymbol{n}_1} + \cdots + a_d \mathcal{D}^{\boldsymbol{n}_d} \in \mathbb{R}\left[\frac{\partial}{\partial x_1}, \cdots, \frac{\partial}{\partial x_m}\right]$ over the ring of partial differential operators (Definition B.3). Now we bound the difference between $\mathcal{T}_g[f]$ and $p(\nabla)[f]$ for every $f \in \mathcal{S}$ and $\boldsymbol{x} \in \mathbb{R}^m$:

$$|\mathcal{T}_g[f](\boldsymbol{x}) - p(\nabla)[f](\boldsymbol{x})| = \left| \int_{\mathcal{W}} \exp(2\pi i \boldsymbol{w}^\top \boldsymbol{x}) \hat{f}(\boldsymbol{w}) \left( \hat{g}(\boldsymbol{w}) - \sum_{k=0}^d a_k q_{\boldsymbol{n}_k}(\boldsymbol{w}) \right) d\boldsymbol{w} \right| \qquad (35)$$

$$\leq \int_{\mathcal{W}} \left| \hat{f}(\boldsymbol{w}) \left( \hat{g}(\boldsymbol{w}) - \sum_{k=0}^d a_k q_{\boldsymbol{n}_k}(\boldsymbol{w}) \right) \right| d\boldsymbol{w} \qquad (36)$$

$$\leq \left( \sup_{\boldsymbol{w} \in \mathcal{W}} \left| \hat{g}(\boldsymbol{w}) - \sum_{k=0}^d a_k q_{\boldsymbol{n}_k}(\boldsymbol{w}) \right| \right) \left( \int_{\mathcal{W}} \left| \hat{f}(\boldsymbol{w}) \right| d\boldsymbol{w} \right) \qquad (37)$$

$$\leq \epsilon, \qquad (38)$$

where Eq. 37 follows from Hölder's inequality, and Eq. 38 is obtained by substituting the upper bound of difference $|\hat{g}(\boldsymbol{w}) - \sum_{k=0}^d a_k q_{\boldsymbol{n}_k}(\boldsymbol{w})|$ and letting $C$ equal to the $L^1$ norm of $\hat{f}(\boldsymbol{w})$ (by Assumption (III)). $\qquad \square$

## C  Implementation Details of INSP-ConvNet

We have formulated exact convolution form and INSP-Conv in Sec. 3.3. We provide a pseudocode to illustrate the forward pass of INSP-ConvNet in Algorithm 1. Below we elaborate each main component:

**Convolutional Layer.**  Each $\mathcal{A}^{(l)}$ represents an implicit convolution layer. We follow the closed-form solution in Eq. 5 to parameterize $\mathcal{A}^{(l)}$ with $\boldsymbol{\theta}^{(l)}$. We point out that $\text{ConvNet}[\Phi]$ also corresponds to a computational graph, which can continuously map coordinates to the output features. To construct this computational graph, we recursively call for gradient networks of the previous layer until the first layer. For example, $\mathcal{A}^{(l)}$ will request the gradient network of $\mathcal{A}^{(l-1)} \cdot \sigma \circ \cdots \circ \mathcal{A}^{(1)} \cdot \Phi$, and then $\mathcal{A}^{(l-1)}$ will request the gradient network of the rest part. This procedure will proceed until the first layer, which directly returns the derivative network of $\Phi$. Kernels in CNNs typically perform multi-channel convolution. However, this is not memory friendly to gradient computing in our framework. To this end, we run channel-wise convolution first and then employ a linear layer to mix channels [90].

**Nonlinear Activation and Normalization.**  Nonlinear activation and normalization are naturally element-wise functions. They are point-wisely applied to the output of an INSP-Net and participate the computational graph construction process. This corresponds to the line 6 of Algorithm 1.

**Training Recipe.**  Given a dataset $\mathcal{D} = \{(\Phi_i, y_i)\}$ with a set of pre-trained INRs $\Phi_i$ and their corresponding labels $y_i$, our goal is to learn a $\text{ConvNet}[\cdot]$ that can process each example. In contrast to standard ConvNets that are designed for grid-based images, the computational graph of INSP-ConvNet contains parameters of both the input INR $\Phi_i$ and learnable kernels $\mathcal{A}^{(l)}$. During the

training stage, we randomly sample a mini-batch $(\Phi_i, y_i)$ from $\mathcal{D}$ to optimize INSP-ConvNet. The corresponding loss will be evaluated according to the network output, and then back-propagate the calculated gradients to the learnable parameters in $\mathcal{A}^{(l)}$, using the stochastic gradient descent optimization. Along the whole process, the parameters of $\Phi_i$ is fixed and only the parameters in $\mathcal{A}^{(l)}$ is optimized. Standard data augmentations are included by default, including rotation, zoom in/out, etc. In practice, we implement these augmentations by using affine transformation on the coordinates of INRs.

## D  Connection with PDE based Signal Processing

Partial Differential Equation (PDE) has been successfully applied to image processing domain as we discussed in Sec. 4.3. In this section, we focus on their connection with our INSP-Net. We summarize the methods of this line of works [61, 40, 66] in the following formulation:

$$\frac{\partial \Psi(\boldsymbol{x}, t)}{\partial t} = M_t \left[ \Psi(\boldsymbol{x}, t), \nabla_{\boldsymbol{x}} \Psi(\boldsymbol{x}, t), \nabla_{\boldsymbol{x}}^2 \Psi(\boldsymbol{x}, t), \cdots \right], \tag{39}$$

where $M_t(\cdot)$ is a time-variant function that remaps the direct output and high-order derivatives of function $\Psi$. For heat diffusion, $M_t$ boils down to be an stationary isotropic combination of second-order derivatives. In [61], $M_t$ is chosen to be a gradient magnitude aware diffusion operator running on divergence operators. [40, 66] degenerate $M_t$ to a time-dependent linear mapping of pre-defined invariants of the maximal order two. We note that Eq. 39 can be naturally solved with INRs, as INRs are amenable to solving complicated differential equation shown by [28]. One straightforward solution is to parameterize $M_t$ by another time-dependent coordinate network [27] and enforce the boundary condition $\Psi(\boldsymbol{x}, 0) = \Phi(\boldsymbol{x})$ and minimize the difference between the two hands of the Eq. 39. However, foreseeable problem falls in sampling inefficiency over the time axis. Suppose we discretize the time axis into small intervals $0 = t_0 < t_1 < \cdots < t_N$, then Eq. 39 has a closed-form solution given $M_t$ by Euler method:

$$\Psi(\boldsymbol{x}, t_{n+1}) = \int_{t_n}^{t_{n+1}} M_t \left[ \Psi(\boldsymbol{x}, t), \nabla_{\boldsymbol{x}} \Psi(\boldsymbol{x}, t), \nabla_{\boldsymbol{x}}^2 \Psi(\boldsymbol{x}, t), \cdots \right] dt + \Psi(\boldsymbol{x}, t_n) \tag{40}$$

$$\approx M_{t_n} \left[ \Psi(\boldsymbol{x}, t_n), \nabla_{\boldsymbol{x}} \Psi(\boldsymbol{x}, t_n), \nabla_{\boldsymbol{x}}^2 \Psi(\boldsymbol{x}, t_n), \cdots \right] (t_{n+1} - t_n) + \Psi(\boldsymbol{x}, t_n). \tag{41}$$

One can see Eq. 41 can be regarded as a special case of our model Eq. 3, where we absorb $M_{t_n}$, time interval $t_{n+1} - t_n$, and the residual term $\Psi(\boldsymbol{x}, t_n)$ into one $\Pi$. Considering our multi-layer model INSP-ConvNet (see Sec. 3.3), we can analogize $t_n$ to the layer number, and then solving Eq. 39 at time $t_N$ is approximately equal to forward passing an $N$-layer INSP-ConvNet.

## E  More Experiment Details

We implement our INSP framework using PyTorch. The gradients are obtained directly using the autograd package from PyTorch. All learnable parameters are trained with AdamW optimizer and a learning rate of 1e-4. For low-level image processing kernels, images are obtained from Set5 dataset [75], Set14 dataset [76], and DIV-2k dataset [77]. These datasets are original collected for super-resolution task, so the images are diverse in style and content. In our experiments, we construct SIREN [28] on each image. For efficiency, we resized the images to $256 \times 256$. We use 90 images to construct the INRs used for training, and use the other images for evaluation.

For image classification, we construct a 2-layer INSP-ConvNet framework. Each INSP layer constructs the derivative computational graphs of the former layers and combines them with learnable $\Pi$. The INSP-layer is capable of approximating a convolution filter. For a fair comparison, we build another 2-layer depthwise convolutional network running on image pixels as the baseline. Both our INSP-ConvNet and the ConvNet running on pixels are trained with the same hyper-parameters. Both experiments take 1000 training epochs, with a learning rate of $1e-4$ using AdamW optimizer.

## F  More Experimental Results

**Additional Visualization.**    In this section, we provide more experimental results. Fig. 9 provides comparisons on edge detection task. Fig. 10 shows image denoising results. Fig. 11 demonstrates

|  | PSNR | SSIM | LPIPS |
|---|---|---|---|
| Input (decoded from INR) | 20.51 | 0.47 | 0.40 |
| MPRNet [81] | 23.95 | 0.72 | 0.36 |
| MAXIM [91] | 24.64 | 0.74 | 0.33 |
| Mean Filter | 22.57 | 0.60 | 0.43 |
| INSP-Net | 23.86 | 0.65 | 0.38 |

Table 2: Quantitative result of image denoising on 100 testing images from DIV-2k dataset [77], where the synthetic noise is rgb gaussian noise. The noise is similar to the ones seen during the training of MPRNet and MAXIM, so they obtain better performance with the help of a much wider training set.

image deblurring results. Fig. 12 shows image blurring results. Fig. 13 shows image inpainting results. Fig. 14 presents additional results on geometry smoothening.

**Additional Quantitative Results.** We also provide quantitative comparisons on the test set in Tab. 2. The test set consists of 100 INRs fitted from 100 images in DIV-2k dataset [77]. In Tab. 2, their performance is better when the synthetic noise becomes three-channel Gaussian noise. The synthetic noise is similar to those seen during the training process of MPRNet [81] and MAXIM [91], so they benefit from their much wider training set.

**Audio Signal Processing.** We additionally validate the ability of our INSP framework by processing audio signals. We add synthetic Gaussian noise onto the audio and use it to fit a SIREN. The noisy audio decoded from the INR is shown in Fig. 15(b). Then we use our INSP-Net to implicitly process it to a new INR that can be further decoded into denoised audio. It's decoded result is shown in Fig. 15(c). We also provide visualization of the denoising effect in Fig. 15(f).

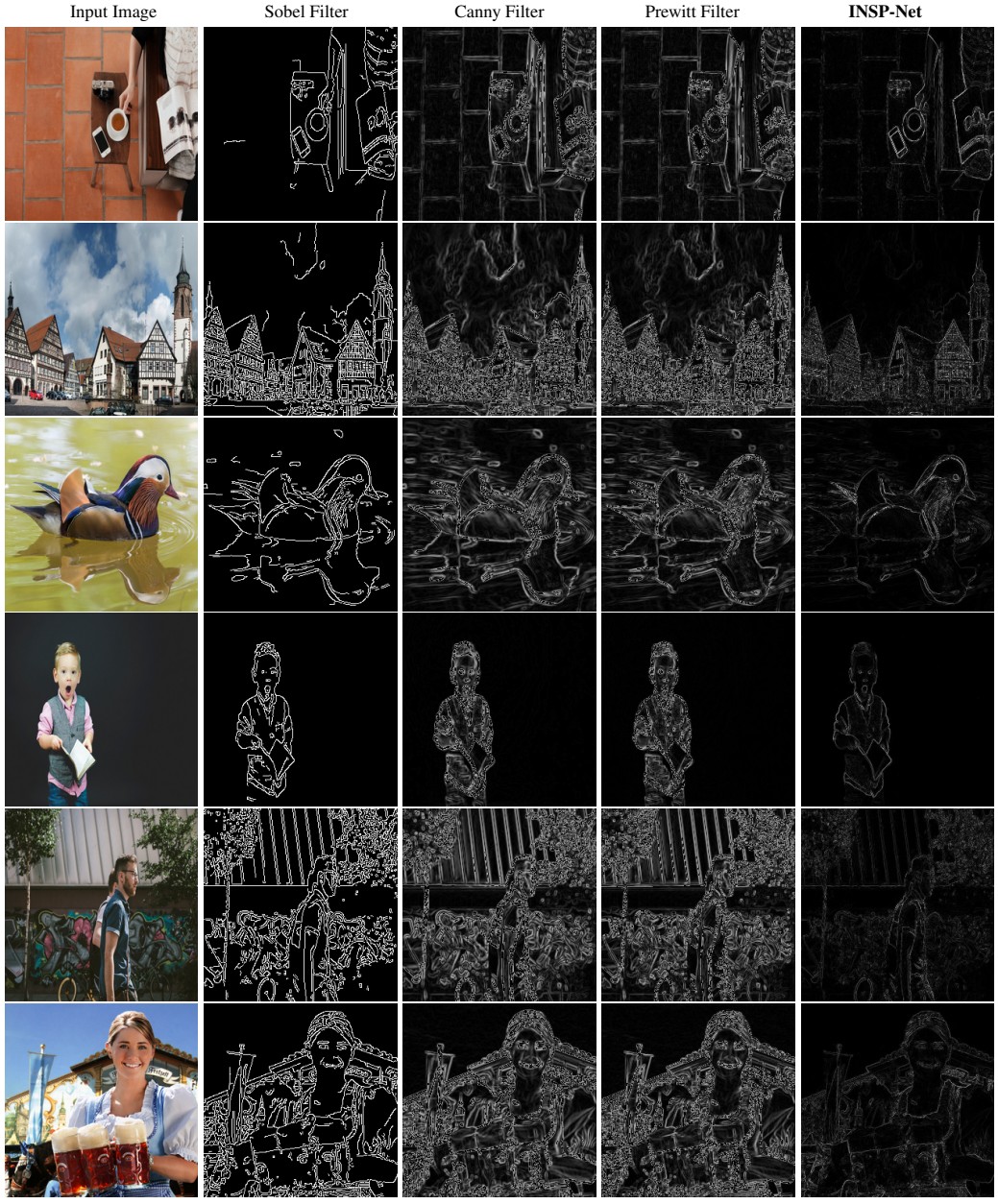

Figure 9: Edge detection. We fit the natural images with SIREN and use our INSP-Net to process implicitly into a new INR that can be decoded into edge maps.

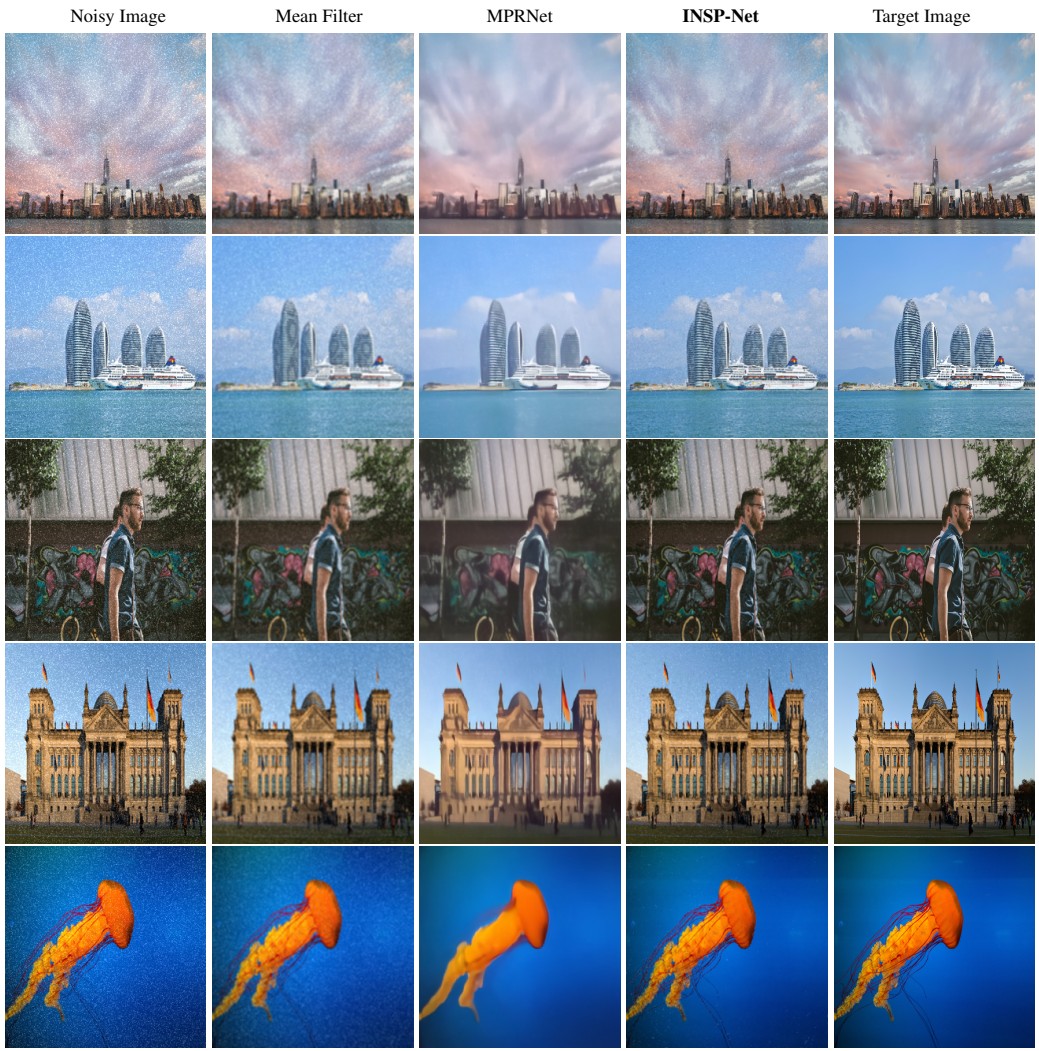

Figure 10: Image denoising. We fit the noisy images with SIREN and train our INSP-Net to process implicitly into a new INR that can be decoded into natural clear images.

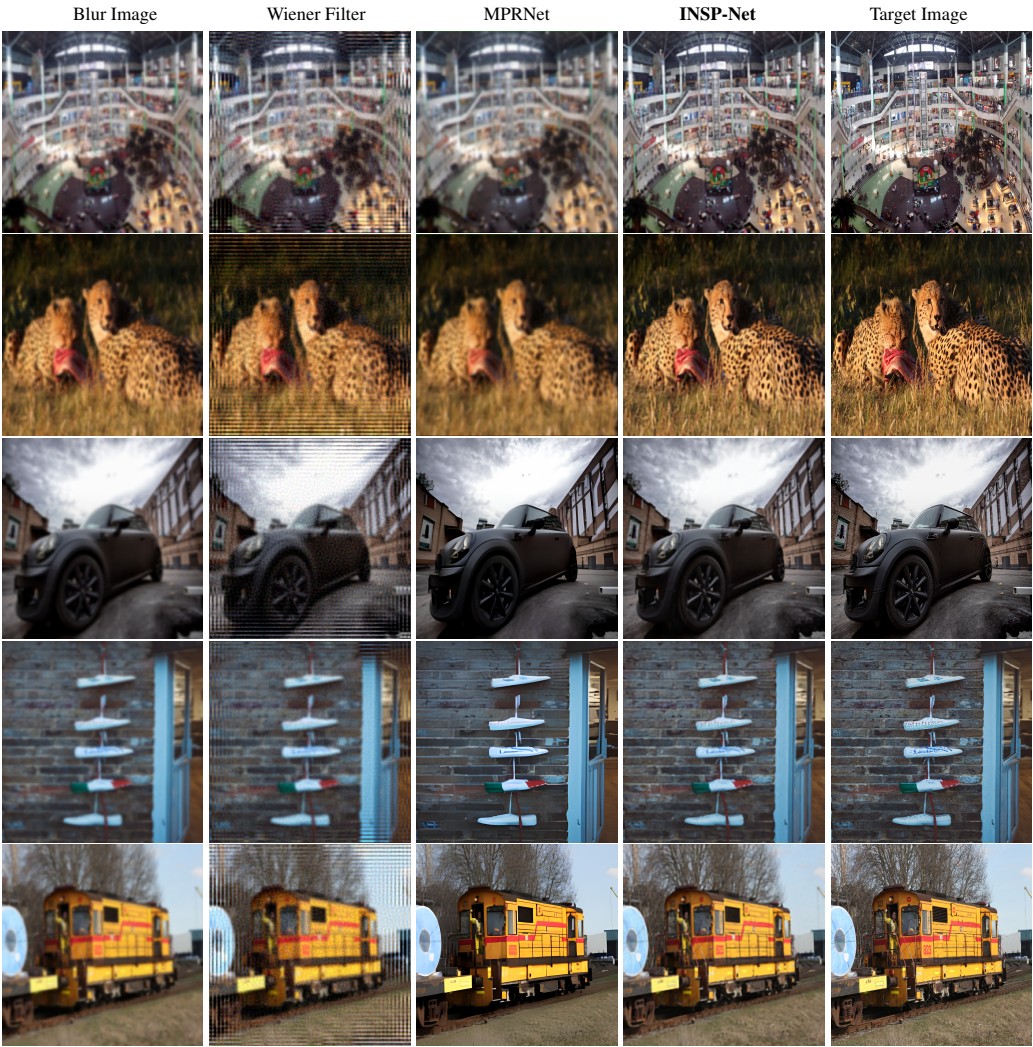

| Blur Image | Wiener Filter | MPRNet | **INSP-Net** | Target Image |

Figure 11: Image deblurring. We fit the blurred images with SIREN and train our INSP-Net to process implicitly into a new INR that can be decoded into clear natural images.

| Original Image | Box Filter | Gaussian Filter | **INSP-Net** |

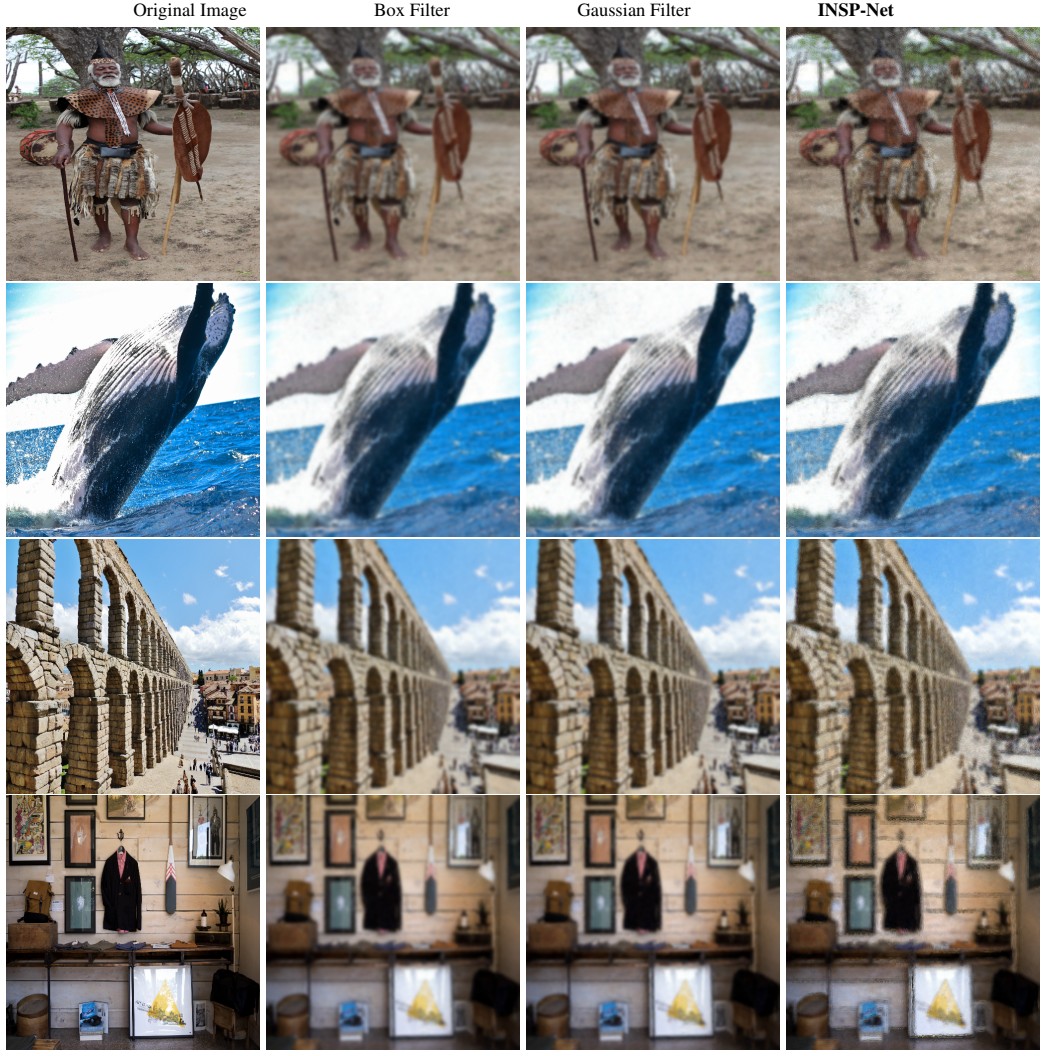

Figure 12: Image blurring. We fit the natural images with SIREN and train our INSP-Net to process implicitly into a new INR that can be decoded into blurred images.

Input Image     Mean Filter     LaMa     **INSP-Net**     Target Image

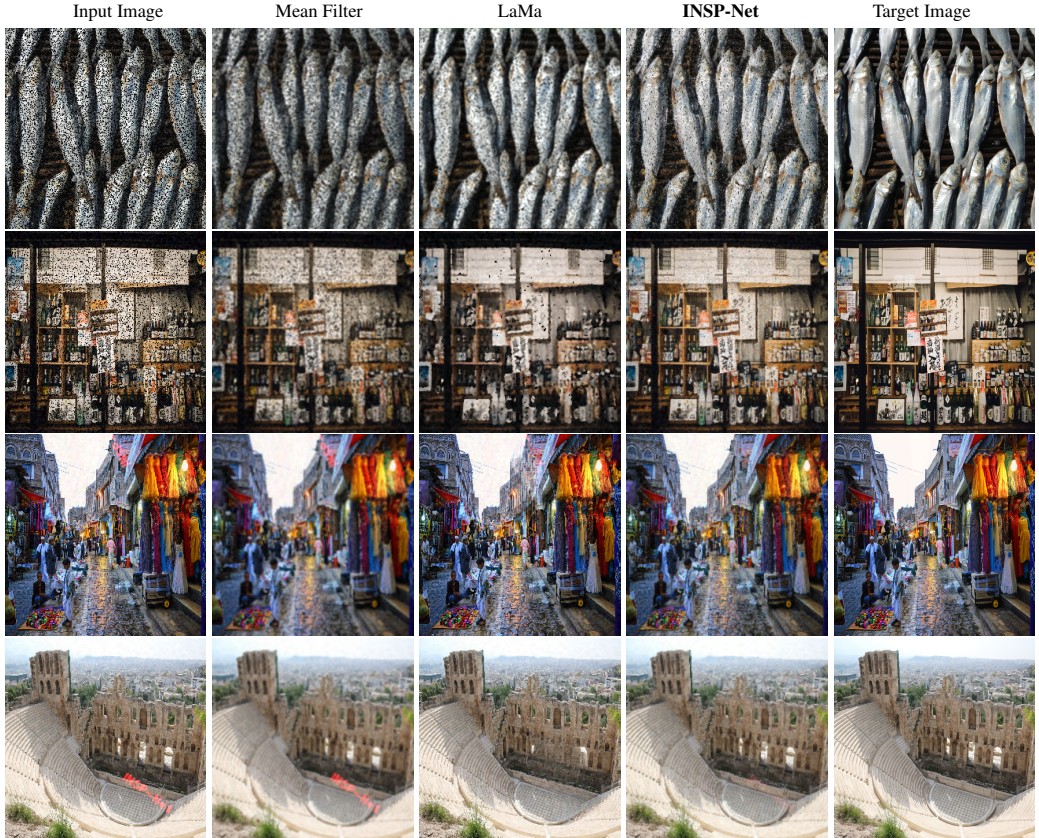

Figure 13: Image inpainting. We fit the input images with SIREN and train our INSP-Net to process implicitly into a new INR that can be decoded into natural images. Note that LaMa requires explicit masks to select the regions for inpainting and the masks are roughly provided. The first two rows contain input images with random pixels erased. The last two rows contain input images with text contamination.

INR Fitted SDF                              SDF Smoothened by INSP-Net

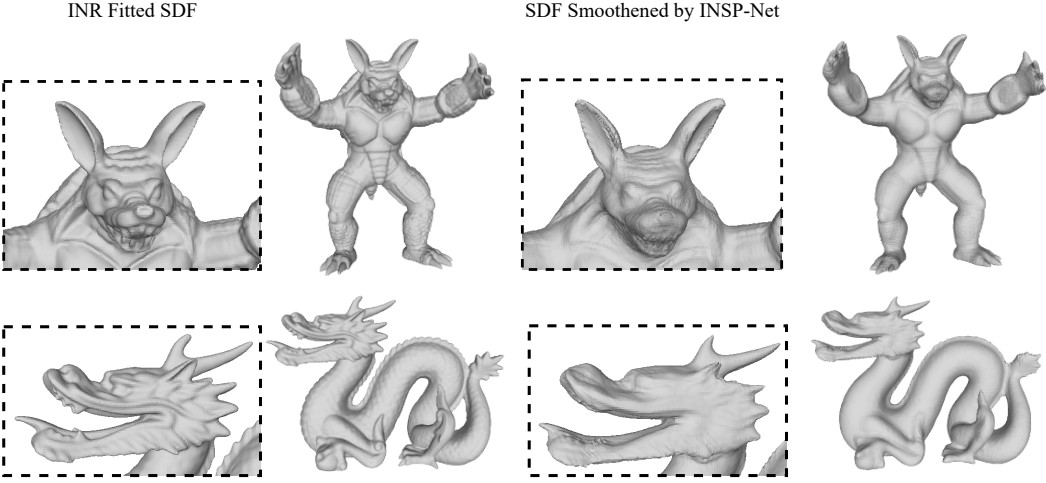

Figure 14: Additional results on geometry smoothening via INSP-Net. Left: unprocessed geometry decoded from an unprocessed INR. Right: smoothened geometry decoded from the output INR of our INSP-Net. Best view in a zoomable electronic copy.

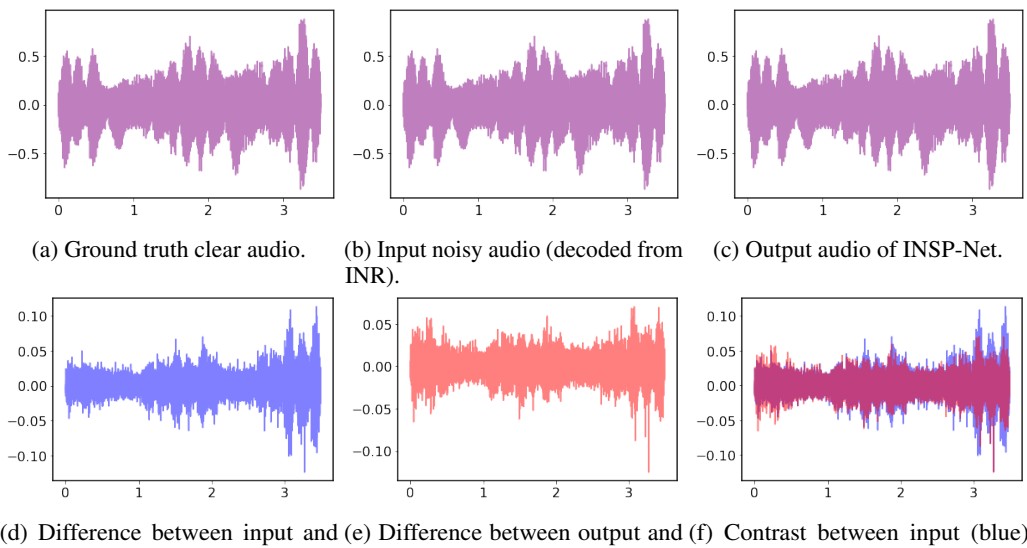

(a) Ground truth clear audio.

(b) Input noisy audio (decoded from INR).

(c) Output audio of INSP-Net.

(d) Difference between input and ground truth.

(e) Difference between output and ground truth.

(f) Contrast between input (blue) and output (red) differences.

Figure 15: Audio denoising. We fit the noisy audio with SIREN and train our INSP-Net to process implicitly into a new INR that can be decoded into denoised audio.