# OpenReview forum: "Signal Processing for Implicit Neural Representations"
_NeurIPS.cc/2022/Conference — NeurIPS 2022 Accept_

### Official Review · Reviewer_qGk1 · 2022-06-27

**Rating:** 5
**Confidence:** 4
**Soundness:** 2 fair
**Presentation:** 3 good
**Contribution:** 3 good

**Summary:**

The paper introduces a novel signal processing framework named INSP-Net that directly operates on INRs without explicit decoding. The INSP-Net can process INRs by combining high-order differential operators. The authors prove any continuous convolution filter can be uniformly approximated by a linear combination of high-order differential operators and empirically validate the effectiveness of the proposed method across both low- and high-level vision tasks.

**Questions:**

See weakness.

**Limitations:**

The authors points out some future works without adequately discussing limitations and negative societal impact.

**Strengths And Weaknesses:**

Strength:

1. The idea is novel. The proposed method attempts to directly process INRs by utilizing high-order differential operators. Theoretically, the authors prove continuous convolution filter can be uniformly approximated by a linear combination of high-order differential operators. Besides, extensive experiments are conducted to validate the effectiveness across various vision tasks.
2. The paper is well organized and easy to be understood.

Weakness:

1. From experiments, it seems that INSP-Net operating on INRs cannot exhibit superiority against convolution operating on pixels. For example, edge detection results of other filters seems contain more detailed textures than that of INSP-Net. In Figures 4 and 5, although the PSNR score is higher, the processed images still contain obvious noise or other artifacts. As for image classification, why is Depthwise CNN used as baseline rather typical CNN?

2. There are some typos in the paper, e.g. Line 77 “sought to to”

---

> ### Author Response · Authors · 2022-08-02
> **Response to Reviewer qGk1**
>
> We thank reviewer qGk1 for the favorable assessments and constructive suggestions. Typos have been fixed in our revision.
>
> **Q1. From experiments, it seems that INSP-Net operating on INRs cannot exhibit superiority against convolution operating on pixels.**
>
> Our experiments are not aimed at achieving state-of-the-art performance on the low-level vision tasks, but to verify the correctness of our theory and the feasibility of our method of directly processing continuous implicit fields on the weight space. Moreover, designing a fair comparison setting can be tricky. As far as we know, a general signal processing framework for implicit fields remains an open question. All the compared baselines are “discrete” methods, and they are served only for reference. We admit the experimental results indicate there is still a gap compared to our final goal, but we believe it is more important to show the value of constructing closed-form operators to edit INR on the weight space, which can potentially lead to a promising direction in the neural implicit field community.
>
> **Q2. For image classification, why is Depthwise CNN used as a baseline rather than typical CNN?**
>
> No existing techniques can classify INR directly using its network weights. Comparing with discrete CNN is for feasibility demonstration and expressive power justification. We use depth-wise CNN because our INSP-ConvNet has the same architecture to trade off the efficiency. As our theory shows, our INSP-ConvNet should have at least the same expressiveness as the compared depth-wise CNN.

---

> ### Author Response · Authors · 2022-08-06
> **Response to Reviewer qGk1**
>
> Dear Reviewer qGk1,
>
> We want to thank you again for your reviewing time and positive assessment. We would like to kindly remind you that the discussion period is ending soon. We hope to use this open response period to discuss the paper to solve the concerns and improve the quality of our paper. Have you gotten a chance to read our responses above, which attempt to address all of your concerns? We would be more than happy to provide more information or clarification.
>
> We have re-stated the significance of our work and demonstrated more experiments on geometry processing (Fig. 8) and audio denoising (Fig. 14). Given all results, for the first time, we demonstrate the possibility of editing INR on the weight and network structure space.
>
> If our response could resolve your concerns, could you please kindly consider raising the rating of our work? We sincerely wish to bring our theoretical grounded solution for direct signal processing on INR into the neural field community.
>
> Best,
>
> Authors of paper 988

---

> > ### Comment · Reviewer_qGk1 · 2022-08-08
> > **After rebuttal**
> >
> > Thanks for providing detailed feedback. All my concerns have been well addressed. I would like to keep my initial rating and make a final rating after discussion with other reviewers.

---

> > > ### Author Response · Authors · 2022-08-08
> > > **Response to Reviewer qGk1**
> > >
> > > We appreciate the reviewer's positive comments. We have updated Fig. 4 which visualizes comparisons against other image denoising methods. We also updated Fig. 8 which shows the application of our model in complex geometry processing. Hope these could strengthen the practical significance of our paper and help this paper turn more positive in your mind.

---

### Official Review · Reviewer_edqT · 2022-06-27

**Rating:** 5
**Confidence:** 4
**Soundness:** 3 good
**Presentation:** 1 poor
**Contribution:** 2 fair

**Summary:**

This paper studies implicit (and continuous) neural representations (INRs) of data defined on discrete lattices. In particular, the authors present a way of performing different operations on these INRs directly without the need for "decoding" them by parametrizing these operators with learnable operators on these representations. The authors demonstrate these ideas on Image denoising, deblurring (and blurring), inpainting and classification.

**Questions:**

- In Eq. (1), did the authors mean \arg\min instead of \arg\max?
- Notation is confusing, as the authors use m and M to denote several things: m is the dimension of the discrete input lattice; e.g. m=2 for images, and M denotes the number of discrete measurements of the continuous function. However, line 117 denotes $x\in R^m$ - shouldn't it be $x\in R^M$? At the same time, Eq. (3) then makes use of $M$ to define the dimension of the domain of the operator $\mathcal A$, which is independent of $m$ and $M$ and depends on the order of the computed derivatives.
- One of the main theses of this paper is that common signal processing operations can be instead defined on INRs, which the authors attempt to demonstrate. However, all of these operators (with the exception of edge detection") are defined only as parametric functions learned from data in a supervised way - as opposed to the original way of defining these operators, "from scratch". This should be clarified, because while gradients can be obtained "in closed form", these operators cannot.
- This reviewer believes that while the idea proposed in this work is nice, the results are far from satisfying:

i) the denoising performance seems quite poor, and the 2 db difference with DnCNN looks very strange given the (inverse) difference in SSIM. Indeed, the INSP-Net image looks considerably more noisy. Moreover, while it's fine comparing with DnCNN, this method is by now 5 years old, and much better algorithm exist (see the methods used for comparison in [1] for a good subset of them). Generally speaking, the denoising performance does not look good and appears far below normal current methods (even below previous methods based on non-local means, dictionary learning, and more).

ii) Likewise, the deblurring performance is also not very satisfying: note the considerable noise introduced in the sky area shown on Fig. 5. The same holds for inpainting (very high levels of noise are introduced in the sky).

iii) Importantly, image processing results are just demonstrations and numbers are presented for only those individual examples - no numbers are reported over the entire dataset.

iv) I also appreciate the comparison on MNIST and CIFAR-10, but such low numbers are concerning. Recall that simple linear models (e.g. PCA +SVM) can better than 87%.

- In line 230, the authors mention that "nonlinear activations and normalizations are naturally element-wise functions", but the authors never state what these are. What are they?
- Throughout the paper, a supervised learning setting is never presented, and so labels are never defined. Thus, it's unclear what the authors mean by "semantic labels y_i" in line 235. Only later in the experimental section one understands what labels the authors are referring to.
- I appreciate the result in Theorem 1, although it's novelty is very limited: indeed, differential operators are shift invariant, and any function defined on norms of vectors is rotation invariant.
- It is unclear why the authors mention that the differential operators form a *wavelet* basis - do these operators satisfy the properties that wavelet must satisfy?
- In line 204, the authors comment that their operators can simulate *exact* convolutions. This seems incorrect to me, as the exact equivalence holds for an infinite sequence, and thus only a finite approximation is possible in this framework.
- Lastly, there are many errors that make reading quite difficult. A non exhaustive list (just the first page) includes:
  - "Implicit Neural Representations (INR) encoding continuous multi-media data..." -> encodes?
  - what are "agnostic" parameters? (line 4)
  - ".. can be computed analytically and invariant to translation" -> invariant to translation, line 11.
  - "presented" -> represented (line 23)
  - "study to encoder" -> study how to encode?
  - what do the authors mean by "parameter stored" in line 33?



[1] Tu, Zhengzhong, et al. "Maxim: Multi-axis mlp for image processing." Proceedings of the IEEE/CVF Conference on Computer Vision and Pattern Recognition. 2022.

**Limitations:**

The authors did not address negative societal impacts nor the limitations of their contribution.

**Strengths And Weaknesses:**

+ This reviewer likes the general idea of this paper. While INR are becoming increasingly popular, it's not obvious how one can perform operations on these continuous elements that are similar to the discrete operations defined over lattices. This paper proposes the first solution to do this (to the best of my knowledge).
+ The method is intuitive and demonstrated on a series of image processing tasks, from low to higher level.

- While the general idea is intuitive, the text is not very clear. There are several typos and general English usage issues that are cumbersome while reading (see below).
- While this demonstrates that these operations can be *approximated* on INRs, the quality of the solutions is not very satisfying. As a result, the advantages of working on the INR domain are not very appealing.

---

> ### Author Response · Authors · 2022-08-02
> **Response to Reviewer edqT**
>
> We thank reviewer edqT for the reviewing time and detailed comments. Typos have been fixed in our revision. However, we are not able to agree with the reviewer's over-concerning about our practical significance due to our experiment results. Our responses to your concerns are listed as follows.
>
> **Q1. The quality of the solutions is not very satisfying. As a result, the advantages of working in the INR domain are not very appealing.**
>
>
> We respectfully disagree that the experimental results will damage the value of this work. INR has achieved undebatable success in signal compression [2], 3D reconstruction [1,3,4], and even scientific computing  [1,5,6]. It enjoys continuous and compact representation, unlimited resolution, and closed-form derivative computation [1]. The ubiquitous application of INR has given us a tantalizing hope of utilizing INR as a universal data format in the future.
>
> The proposed framework is the first generic framework that processes INRs without breaking down their implicit and continuous nature by leveraging closed-form differential operators. Although the experimental results are not demonstrated to show superiority in every aspect (e.g., it is comparably slow compared to traditional filters), they are still encouraging, considering that the progress of INR still lies in an emerging stage (In [7], authors revealed the imperfectness of existing INRs). Moreover, frankly speaking, finding comparable baselines is tricky since all the existing image processing methods are established for vectorized signals. That being said, all the baselines are processing totally different input data from our method. Thus they are not fairly compared. Even so, we have successfully demonstrated the promise of adopting continuous convolution on INR, which largely outperforms discrete convolution-based methods.
>
> The performance of our method is close to the state-of-the-art "discrete" models. Even though it indicates there is still a gap compared to our final goal, we believe it is more important that this work will potentially motivate following future works and shed light on how to directly process INR without decoding it. As admitted by Reviewer 1Ys6,  our work makes the first step towards analytically processing implicit fields, and can "inspire work that will push this idea further, maybe eventually to the point where we will be able to compute functions of neural fields directly in the weight space".
>
> As per reviewer's request, we also list the average numerical results on the full test set in Tab. 2 and Tab. 3. Additionally, we create two baselines on image classification on MNIST dataset: directly applying 1) MLP and 2) PCA + SVM on the INR weights. Their inferior performances (9.8% for MLP and 11.3% for PCA + SVM) further support the significance of our algorithm.
>
>
> Table 2: Quantitative result of image denoising on 100 testing images from DIV-2k dataset, where the synthetic noise is gray gaussian noise clipped to be positive. MPRNet and MAXIM overfit to the SIDD dataset they were trained on, and don't generalize well on the synthetic Gaussian noise we are using here.
>
> |                          |    PSNR   |   SSIM   |   LPIPS  |
> |:------------------------:|:---------:|:--------:|:--------:|
> | Input (decoded from INR) |   23.23   |   0.64   |   0.33   |
> |          MPRNet          |   21.70   |   0.73   |   0.32   |
> |           MAXIM          |   24.32   |   0.80   |   0.26   |
> |         INSP-Net         | **27.95** | **0.82** | **0.23** |
>
> Table 3: Quantitative result of image denoising on 100 testing images from DIV-2k dataset, where the synthetic noise is rgb gaussian noise. The noise is similar to the ones seen during the training of MPRNet and MAXIM, so they obtain better performance with the help of a much wider training set.
>
> |                          |  PSNR | SSIM | LPIPS |
> |:------------------------:|:-----:|:----:|:-----:|
> | Input (decoded from INR) | 20.51 | 0.47 |  0.40 |
> |          MPRNet          | 23.95 | 0.72 |  0.36 |
> |           MAXIM          | 24.64 | 0.74 |  0.33 |
> | Mean Filter | 22.57 |  0.60 |  0.43 |
> |         INSP-Net         | 23.86 | 0.65 |  0.38 |
>
> [1] Sitzmann et al. Implicit Neural Representations with Periodic Activation Functions
>
> [2] Dupont et al. COIN: COmpression with Implicit Neural representations
>
> [3] Park et al. DeepSDF: Learning Continuous Signed Distance Functions for Shape Representation
>
> [4] Mildenhall et al. NeRF: Representing Scenes as Neural Radiance Fields for View Synthesis
>
> [5] Li et al. Fourier Neural Operator for Parametric Partial Differential Equations
>
> [6] Zhong et al. CryoDRGN: reconstruction of heterogeneous cryo-EM structures using neural networks
>
> [7] Yuce et al. A Structured Dictionary Perspective on Implicit Neural Representations

---

> > ### Author Response · Authors · 2022-08-02
> > **Response to Reviewer edqT**
> >
> > **Q2. Notation confusion.**
> >
> > In Eq. 1, indeed we meant $\arg\min$ instead of $\arg\max$. We have fixed this typo in our revision. To avoid confusion, we have also replaced the notation of number of measurements in Eq. 2 to $N$. We used $m$ to denote the dimension of input coordinate, $M$ to denote the input dimension of $\Pi$. Hence, we still believe in line 117, it should be $x \in R^m$ instead of $R^M$.
> >
> >
> > **Q3. Gradients can be obtained "in closed form", but operators defined only as parametric functions learned from data operators cannot.**
> >
> > By saying "in closed form", we mean the forward pass of processed INR is a mathematical expression with a finite number of standard operations and can be computed analytically without any discretization and approximation, instead of meaning computed the parameterization is a closed form solution to some objectives. AutoInt [1] used similar terminology. We added a footnote to clarify this definition.
> >
> > [1] Lindell et al. AutoInt: Automatic Integration for Fast Neural Volume Rendering
> >
> >
> > **Q4. How to handle nonlinear activations and normalizations?**
> >
> > Since nonlinear activation and normalization are usually element-wise operations, we can directly compose such operations to the output of INSP-Net (Eq. 3). To illustrate how convolutional layers and nonlinearity cooperate with each other, we include an algorithm paradigm in Algorithm 1.
> >
> > **Q5. The novelty of Theorem 1 is very limited.**
> >
> > We are aware that group invariance has been well investigated in partial differential equations (line 148-152), and we also cite a series of those papers. Even though Theorem 1 is not our main result, we still feel it is necessary to formalize it as a theorem in our paper, which highlights some desirable properties of our framework to the readers.
> >
> > **Q6. It is unclear why the authors mention that the differential operators form a wavelet basis.**
> >
> > Differential operators are not strictly orthogonal to each other, however, they form an independent basis. We have removed the word "wavelet" in the revision to avoid confusion.
> >
> > **Q7. In line 204, the authors comment that their operators can simulate exact convolutions. This seems incorrect to me, as the exact equivalence holds for an infinite sequence, and thus only a finite approximation is possible in this framework.**
> >
> > We respectfully disagree with the reviewer's opinion on our Eq. 5 is not an exact convolution. Let's recall that convolution is defined as the linear and shift-invariant operator. As we revealed in Thm. 1, Eq. 5 is both linear and shift-invariant, thus simulating an exact convolution. It is safe to state $A$ is approximating any convolution inside the linear and shift-invariant family when fixing $\Pi$ to be a linear transformation (cf. line 207-208).
> >
> > **Q8. There are many errors that make reading quite difficult. It's unclear what the authors mean by "semantic labels y_i" in line 235.**
> >
> > Thanks for pointing out writing errors. We have fixed all the mentioned typos and proofread the main text.
> >
> > **Q9. What are "agnostic" parameters? (line 4) What do the authors mean by "parameter stored" in line 33?**
> >
> > "Agnostic parameters" means the INR's parameters in the weight space have no intuitive meaning, being agnostic to people. We replace this terminology with "unintuitive parameter" in our revision. "Parameter stored" means all the parameters of a model which store all the information for the represented signal.

---

> > > ### Comment · Reviewer_edqT · 2022-08-06
> > > **Response to the authors**
> > >
> > > I thank the authors for their thorough responses to my questions and comments. Some follow ups:
> > >
> > > Q1. I completely understand that comparing is hard, given the discrete vs continuous nature of the different algorithms. Yet, the authors study image processing tasks, and images and signals are (for most practical purposes) discrete objects, so this is inevitable. I still believe the denoising performance (as well as in other tasks) if far behind conventional methods. Once again, I don't understand how a difference of 2dB (w.r.t DnCNN) is possible in the example on Fig. 4, given that the results of INSP-Net is so much more noisy.
> > >
> > > I certainly appreciate the inclusion of the Tables and the comparison with other 2 methods for image denoising, and they indeed improve and clarity the contribution of this paper. One small comment: The authors can include Table 2 (denoising on noise that is not Gaussian) but this does not mean much: it is clear that these off-the-shelf denoising models were trained on a specific data distribution (in this case, natural images + Gaussian noise). These methods performing bad under different distributions has nothing to do with overfitting (which implies fitting to a finite sample *from the same distribution as the testing data*), but rather with how robust these are to shifts in the data distribution. I presume the authors did not make any claims on robustness to distribution shifts, and as a result the comparison seems unfair. The authors may choose to keep this table if they wish - but I would remove it if this were my paper :)
> > >
> > > Q2. Thanks
> > >
> > > Q3. Thanks for the clarification: indeed, I don't think this is a very standard definition of "closed form".
> > >
> > > Q4. Thanks.
> > >
> > > Q5. Thanks
> > >
> > > Q6. Thanks. As a remark: wavelet bases and an (orthogonal) basis are very different things. One can indeed have wavelet frames that are not orthogonal.
> > >
> > > Q7. Thanks for the clarification. Yet, this pointed to a new question on the same topic: Theorem 1 states that "$\mathcal A$ can be shift invariant *for every* $\Pi$". I agree with the claim that "every shift invariant operator can be represented by *some* $\mathcal A$", but this is not what the statement of the theorem says. My concern comes from the following observation: Consider the operator $\Pi$ given by $\Pi=\Phi(x)$, which is certainly allowed by the definition in Eq. (3). Now, in this case, $\Pi$ will not be shift invariant (unless $\Phi(x)$ represents a constant image...), which contradicts the original statement. What am I missing?
> > >
> > > Q8. Thanks.
> > >
> > > Q9. Thanks.

---

> > > > ### Author Response · Authors · 2022-08-08
> > > > **Response to Reviewer edqT**
> > > >
> > > > We thank the reviewer for their insightful comments. Our responses to the concerns are listed as follows.
> > > >
> > > > **Q1. The quality of the solutions is not very satisfying. As a result, the advantages of working in the INR domain are not very appealing.**
> > > >
> > > > We appreciate the reviewer’s comments on Fig. 4 and Tab. 2. Though the result of DnCNN has a better visual looking, the result of our INSP-Net has a smaller pixel-wise error against the reference image. The reason why the quantitative performance of DnCNN/MPRNet is lower is that they obtain a blur result. We’ve updated Fig. 4 with Gaussian noise examples and removed the original Tab. 2 from the pdf as suggested by the reviewer.
> > > >
> > > > We would also like to re-emphasize the advantages of INRs as well as our signal processing framework.
> > > >
> > > > 1. INR has demonstrated great success in image and video compression [1,2]. These exciting advances advocate INR to potentially become the next universal data compression and representation format for the future.
> > > >
> > > >     There already exist various attempts to process image [4] and video [3] compressed representation directly, alleviating the unnecessary cost of decoding the representations first. Thus, a corresponding continuous processing method for INRs without the need of decoding is in great demand.
> > > >
> > > > 2. INRs have now become irreplaceable in many computer vision tasks, such as reconstructing SDF. As shown by [5], only adopting INR can effectively solve the Eikonal boundary value problem to recover fine details in 3D geometries. In this scenario, any attempt of discretization will cause information loss. Our method directly edits SDF on the network weights and structures, which preserves all the information in the continuous space. We updated our results in Fig. 8, which demonstrates the capability of our INSP-Net in processing complex geometries encoded by INRs.
> > > >
> > > > Our work on continuous signal processing is forward-looking. We do not expect to outperform existing endeavors for decades in processing the discretized instances. However, our novel setting is highly worth investigating given its above-mentioned advantages.
> > > >
> > > >
> > > > [1] Zhang et al. Implicit Neural Video Compression
> > > >
> > > > [2] Dupont et al. COIN: COmpression with Implicit Neural representations
> > > >
> > > > [3] Wu et al. Compressed Video Action Recognition
> > > >
> > > > [4] Xu et al. Learning in the Frequency Domain
> > > >
> > > > [5] Sitzmann et al. Implicit Neural Representations with Periodic Activation Functions
> > > >
> > > > **Q7. Given by $\Pi = \Phi(x)$, $\Pi$ will not be shift-invariant (unless $\Phi$ represents a constant image...).**
> > > >
> > > > First of all, we would like to remind the reviewer of our notation. $\mathcal{A}$ is the operator (e.g., a convolution) that maps a function (INR) to another function (INR), while $\Pi$ as a part of $\mathcal{A}$ is just a multivariate function. Operator $\mathcal{A}$ maps a function by composing $\Pi$ on top of its derivatives. An operator is shift-invariance if it is commutative with coordinate transformation (cf. Definition 1 & 2). **Our Theorem 1 discusses the invariance of $\mathcal{A}$ instead of $\Pi$.** $\mathcal{A}$ is shift-invariant regardless of whether $\Pi$ is shift-invariant because $\Pi$ is a pointwise function with respect only to derivatives (which are known to be shift-invariant).
> > > >
> > > > We do not think $\Pi = \Phi(x)$ is generally allowed since their input spaces are different (The input space of $\Pi$ is $M$-dimension while $\Phi$ is $m$-dimension). So we guess the reviewer means $\mathcal{A}\Phi(x) = \Phi(x)$, i.e., $\mathcal{A}$ is an identity mapping. Clearly, the identity operator is shift-invariant since $ \mathcal{A}[\Phi \circ T] (x) = \Phi \circ T (x) = \mathcal{A}[\Phi] \circ T (x)$. We have to admit that rigorously speaking, we need to change “invariance” to “equivariance”. However, since “invariance” is a common “misnomer” used over decades in computer vision [1], we do not see there is a risk of causing confusion in this community.
> > > >
> > > > [1] Lecun et al., Gradient Based Learning Applied to Document Recognition, 1998

---

> > > > > ### Comment · Reviewer_edqT · 2022-08-09
> > > > > **Response to Authors**
> > > > >
> > > > > I appreciate the responses from the authors. I'm increasing my score to 5 at this point, while I look forward to discussing more with the other reviewers in the next phase.

---

> ### Author Response · Authors · 2022-08-06
> **Response to Reviewer edqT**
>
> Dear Reviewer edqT,
>
> We want to thank you for the constructive comments in your review. As a follow-up on our responses, we would like to kindly remind you that the discussion period is ending soon. We hope to use this open response period to discuss the paper to solve the concerns and improve the quality of our paper. Have you gotten a chance to read our responses above, which attempt to address all of your concerns?
>
> We have fixed all the writing flaws and provided more experiments on geometry processing (Fig. 8) and audio denoising (Fig. 14). Given all preliminary results, for the first time, we demonstrate the possibility of editing INR on the weight and network structure space.
>
> We sincerely hope to have further discussions with you to see if our response solves the concerns. We would be more than happy to provide more information or clarification, should it be necessary. We hope our paper could receive a positive and fair assessment. We sincerely wish to bring our theoretical grounded solution for direct signal processing on INR into the neural field community.
>
> Best,
>
> Authors of paper 988

---

### Official Review · Reviewer_Uh51 · 2022-07-06

**Rating:** 6
**Confidence:** 5
**Soundness:** 3 good
**Presentation:** 2 fair
**Contribution:** 2 fair

**Summary:**

Motivated by recent advances in implicit neural representation (INR) for images, this paper seeks to develop an approach to modify learned INR *"without decoding”*. Specifically, the paper proposes training an additional network which operates on the high-order spatial derivatives of the original INR previously fitted for some image. This extra neural network takes in the original INR prediction, as well as its high-order derivatives w.r.t. the input coordinates (hence the spatial derivative); the objective of this network depends on the specific task, and the paper discusses tasks such as denoising, deblurring, inpainting, and classification.

**Questions:**

* Why is the title about general signal processing but the paper only deals with images?
* What happened to the geometric processing and the 3D shape shown in Fig. 1?
* For Fig. 4 to 7, why not show the initial decoded images after fitting the INR on the corrupted images? How much does the extra network actually help?
* Isn’t the INSP-Net with the “A” kernel just a fully connected MLP (Line 119)? Why would stacking multiple INSP-Net be called INSP-ConvNet? Is it because MLP is equivalent to 1D Convolution? By this logic, isn’t the original INR also a “ConvNet”?
* What does “modify an INR without explicit decoding” mean? Doesn’t the proposed method need to decode the INR (Eq. 3) by evaluating the “acquired INR” on every single pixel “x”? Isn’t that decoding already?
* If we need to evaluate every pixel “x” with the INR, before feeding it to the extra network, what benefit does it bring over simply using the decoded image?
* Why not show the inference speed / network run-time? How much slower would it be due to computing the high-order derivatives?

In particular, it would be critical to elaborate the "without decoding" claim, clarify why the proposed method is not also decoding from the INR, and how the proposed INSP-ConvNet really is **convolutional** instead of just **multiple MLPs**. These would determine the validity of the main claims and conceptual contributions of this paper, no matter how good or bad the experimental results are.

**Limitations:**

The authors have not really addressed them. Discussion about at least the limitation of a potentially slower run-time should be expected in the revision.

**Strengths And Weaknesses:**

Strengths:

**Clear and reasonable motivations.**
INR has achieved great success in accurately fitting images, and it is therefore well-motivated to explore possibly modifying the learned INR for further downstream tasks. It is also admirable that the paper points out the potential benefits of improving INR based on ideas from the field of digital signal processing

**Variety of tasks.**
This paper presents a wide range of applications, such as edge detection, blurring, deblurring, denoising, inpainting, and classification. It is helpful for the community to be aware of the huge potential of INR of images.

Weaknesses:

**Experimental design.**

* The title suggests a general framework of signal processing, and the teaser figure (Fig. 1) shows a 3D mesh and explicitly states “geometric processing”, but the paper is actually only about image processing.

* The paper has no experiment on geometric processing on signed distance fields, despite the fact that DeepSDF [10] is one of the most prominent examples of successful INR applications.

* The paper has no experiment on signal processing of time series signals (e.g. audio or speech), which is arguably the most representative data domain for the entire field of digital signal processing. The SIREN paper [27] that this paper is based on has covered audio signals, so it is not a foreign type of data for INR.

* It is completely fine if a paper only focuses on images, but the title and its first figure should at least appropriately reflect the content and scope.

**Validation rigor.**
* The paper does indeed cover a lot of image processing applications, but it lacks sufficient rigor in validating the proposed approach.
The paper presents quantitative metrics (PSNR/SSIM) for five selected images only (Figures 4 to 7). Those provided in the supplement do not contain any quantitative metric, and even if they do, the paper should still provide average results across a large dataset to avoid cherry picking.
* In the denoising/deblurring/inpainting experiments, the INR is first fitted against the noisy/blurry/masked image before then processed by the task-specific network. The results seem to suggest that this task-specific network can successfully denoise/deblur/inpaint from the corrupted input image. However, the paper does not show the decoded image from the original INR. It is well-known that INR itself can at least do denoising [30], so it is not clear how much the proposed training actually improves from the INR simply fitted against the corrupted input image.
* In the classification experiment, it is not clear how only training a 2-layer depthwise CNN is a “fair comparison”. The paper does not provide the number of network parameters, nor does it explain why it does not consider training a small MLP.
* Across all experiments, it is not clear what number k, or how many high-order derivatives, should one use. The paper presents no analysis on the impact of having more high-order derivatives. It is not even clear if one really needs high-order derivatives.

**Literature review.** \
*Some related papers are not correctly interpreted.*
* While discussing “existing approaches" for “editing on INRs”, the paper mentions (Line 36 to 37) that “another main direction” and only references three works, [27], [28], and [29]. However, [28] and [29] are both from 2017, well before the recent introduction of INR, whose main characteristic is the “coordinate-to-value mapping” (Line 25). In fact, [28] makes no use of neural networks, and does not even mention the phrase “neural network” at all; [29] is about using GAN to generate 3D voxels, hardly related to INR either.
* While introducing FFM (Line 89 to 98) and Eq. (2), the notation seems to suggest that “positional encoding” is merely a sine function (Line 97 to 98), and ignores the fact that FFM [30] introduces the Gaussian mapping and uses it by default.
* Section 4.2 discusses the supposedly related works on “editing the reconstructed 3D scenes”. This paper only focuses on image processing, so the relevance of this subsection is really weak.

*Some highly related papers are not included.*
* Using the derivatives of INR has been explored by [a] (with similar computational graph discussion) in the context of faster INR-based rendering.
* Using INR for downstream applications (e.g. classification) has been explored by [b].
* Using INR for image and video compression has been also explored by [c] (with a PDE-based positional encoding) in the context of light fields.
* [d] introduces a technique that significantly accelerates the fitting of INR on 2D images.
* [e] introduces a technique for improving the representation efficiency of INR of images via reducing the parameter count.

* These works contain far higher relevance to this paper, compared to the 7 referenced papers on 3D shapes (there’s no 3D shape in this paper) and the other 7 referenced papers about 3D GANs

[a] Autoint: Automatic integration for fast neural volume rendering. Lindell et al. CVPR 2021. \
[b] From data to functa. Dupoint et al. ICML 2022. \
[c] Signet: Efficient Neural Representation for Light Fields. Feng et al. ICCV 2021. \
[d] Learned Initializations for Optimizing Coordinate-Based Neural Representations. Tancik et al. CVPR 2021. \
[e] Meta-learning Sparse Implicit Neural Representations" Lee et al. NeurIPS 2021.

---

> ### Author Response · Authors · 2022-08-02
> **Response to Reviewer Uh51**
>
> We thank reviewer Uh51 for the time and actionable suggestions on providing more demonstration beyond the image domain. However, especially for your concerns on our experimental soundness, we would emphasize that our experiments are not aimed at showing the state-of-the-art performance on the low-level vision tasks, but to verify the feasibility of directly processing continuous implicit fields without explicit decoding. Actually, designing a fair comparison setting can be tricky. All the compared baselines are “discrete” methods. To our best knowledge, there is no prior method that can edit INR in the continuous regime. That being said, the input data of our methods and compared baselines are totally different. For other questions, we also provide detailed responses as below.
>
> **Q1. The paper is actually only about image processing. And more experiments on geometric processing and time-series signal processing.**
>
> Thanks for proposing to apply our method to other representative data domains. Our method is a generic framework for processing all multimedia signals. We proved that our framework is expressive enough to represent continuous convolution. Therefore, we can safely believe the application range of INSP-Net is at least as wide as the convolution based signal processing techniques. Our method should be independent of the data modality and can be easily extended to other data. To support our argument, we have provided the additional results in geometry smoothing and audio denoising in our revision. Please refer to Fig. 8 and Fig. 14. Our new results demonstrate the feasibility of our INSP-Net in 3D and audio domains.
>
> **Q2. The paper should provide average results across a large dataset to avoid cherry-picking.**
>
> The numbers listed in the paper are only given to quantitatively demonstrate our method’s feasibility. And the originally shown examples are non-curated. As per reviewer’s request, we have included a table of average scores evaluated on a large dataset in our revision. Please refer to Tab. 2 and Tab. 3.
>
>
> Table 2: Quantitative result of image denoising on 100 testing images from DIV-2k dataset, where the synthetic noise is gray gaussian noise clipped to be positive. MPRNet and MAXIM overfit to the SIDD dataset they were trained on, and don’t generalize well on the synthetic Gaussian noise we are using here.
>
> |                          |    PSNR   |   SSIM   |   LPIPS  |
> |:------------------------:|:---------:|:--------:|:--------:|
> | Input (decoded from INR) |   23.23   |   0.64   |   0.33   |
> |          MPRNet          |   21.70   |   0.73   |   0.32   |
> |           MAXIM          |   24.32   |   0.80   |   0.26   |
> |         INSP-Net         | **27.95** | **0.82** | **0.23** |
>
> Table 3: Quantitative result of image denoising on 100 testing images from DIV-2k dataset, where the synthetic noise is rgb gaussian noise. The noise is similar to the ones seen during the training of MPRNet and MAXIM, so they obtain better performance with the help of a much wider training set.
>
> |                          |  PSNR | SSIM | LPIPS |
> |:------------------------:|:-----:|:----:|:-----:|
> | Input (decoded from INR) | 20.51 | 0.47 |  0.40 |
> |          MPRNet          | 23.95 | 0.72 |  0.36 |
> |           MAXIM          | 24.64 | 0.74 |  0.33 |
> |         INSP-Net         | 23.86 | 0.65 |  0.38 |
>
>
> **Q3. It is not clear how much the proposed training actually improves from the INR simply fitted against the corrupted input image. Why not show the initial decoded images after fitting the INR on the corrupted images?**
>
> We observe that INR is not naturally against all corruptions. The unprocessed input images shown in the original draft are already the decoded images from the original INR. We’ve updated the wording to avoid confusion.
>
> **Q4. How only training a 2-layer depthwise CNN is a “fair comparison”. Why does it not consider training a small MLP?**
>
> We only intend to verify the learning expressiveness of our framework by showing comparable results with CNNs on image classification. We choose the two-layer depthwise CNN because we believe our two-layer INSP-ConvNet is mathematically as powerful as this two-layer depthwise CNN. The purpose of comparing the number of parameters remains unknown and thus, its conclusions may be less meaningful. CNN and INSP-ConvNet are processing different data. Our INSP-ConvNet is learning on continuous and implicit signals (or namely, a set of MLP networks), which presumably requires a larger parameter budget.
> We provide additional baselines on MNIST dataset that process the weight of the INR directly: 1) We train an MLP classifier with the same number of parameters as our INSP-ConvNet that takes the MLP weights as input, its classification accuracy is 9.8%; 2) We use PCA + SVM to classify the weight of the INRs, its classification accuracy is 11.3%.

---

> > ### Author Response · Authors · 2022-08-02
> > **Response to Reviewer Uh51**
> >
> >
> > **Q5. Across all experiments, it is not clear what number k, or how many high-order derivatives, should one use.**
> >
> > For image processing, we set k=5. For geometric processing, we set k=4. For audio denoising, we set k=3. The k values are empirically set to fit in the GPU memory and shared across all corresponding experiments, without any bell-and-whistle tuning per case.
> >
> > **Q6. Some related papers are not correctly interpreted or not included.**
> >
> > Thanks for pointing out the incorrect interpretation of existing works for INR editing. We have revised this section according to the reviewer’s suggestions. For FFM [30], to our best understanding, the Gaussian mapping mentioned in their paper is just another variant of Fourier feature mapping. It simply samples the frequency weights from a Gaussian distribution, which does not contradict our description “Fourier Feature Mapping (FFM) places a sinusoidal transformation before the MLP”. We still insist Sec. 4.2 is necessary, because it shows the demand and promise of editing implicit fields. As we added a demonstration for 3D object editing, discussing them is no more out-of-scope. We also added the missing citations mentioned by the reviewer.
> >
> > **Q7. Isn’t the INSP-Net with the “A” kernel just a fully connected MLP (Line 119)? Why would stacking multiple INSP-Net be called INSP-ConvNet?**
> >
> > The INSP-Net is not only a fully-connected MLP, but also includes all the computation steps of derivatives. By simplifying the MLP $\Pi$ to a linear layer (Eq. 5), INSP-Net can precisely simulate a convolution operator (i.e., a conv layer). By interleaving Eq. 5 with non-linearity, we form a multi-layer structure, and we call it INSP-ConvNet because it simulates “continuous” CNN on INRs.
> >
> > **Q8. What does “modify an INR without explicit decoding” mean?**
> >
> > “Modification without explicit decoding” means we do not need to query the INR at every grid point and extract it to a pixel array before processing it. Recall our formulation Eq. 3, to compute the value at point $x$, INSP-Net only does pointwise computation for $x$ by feeding different order derivatives at $x$ into the MLP $\Pi$ instead of the derivatives at every location $x$. Eq. 3 as a whole is a computational graph that can output the value at arbitrary coordinates without explicitly decoding other points, which therefore can be regarded as the network representing the processed INR. We decode Eq. 3 to image grids only when visualizing or evaluating its visual quality.
> >
> > **Q9. Why not show the inference speed / network run-time? How much slower would it be due to computing the high-order derivatives?**
> >
> > Taking the audio denoising task as an example, the original unprocessed INR takes on average 0.01s to inference a 3.5s audio, while ours takes 0.12s (12x). We have added a limitation paragraph in our revision to clarify the efficiency problem. We leave more efficient processing operator representation for future works. We recall that even a plain INR is much slower than “explicit” methods. Likewise, processing continuous representation should presumably be much heavier than processing on regular grids. Even so, directly operating INR on the weight space is always a desirable feature since it does not involve discretization or approximation, which may cause information loss.

---

> ### Author Response · Authors · 2022-08-06
> **Response to Reviewer Uh51**
>
> Dear Reviewer Uh51,
>
> We appreciate it if you could please review our response to your comments since the reviewer-author discussion session has started several days. We have provided more results on geometry smoothing (Fig. 8) and audio denoising (Fig. 14).  We are willing to reply to your further questions in this time window if you have any feedback.
>
> If our response could resolve your concerns, could you please kindly consider raising the rating of our work? We sincerely wish to bring our theoretical grounded solution for direct signal processing on INR into the neural field community. Thank you very much for your efforts and time.
>
> Best,
>
> Paper 988 Authors

---

> > ### Comment · Reviewer_Uh51 · 2022-08-07
> > **Response**
> >
> > Thank you for the detailed response. I appreciate the additional information regarding geometry and audio processing, which was originally missing from the paper. I would be open to increase the rating, after further discussions and confirmations with other reviewers.

---

> > > ### Author Response · Authors · 2022-08-08
> > > **Response to Reviewer Uh51**
> > >
> > > We thank the reviewer for their positive comments. We've updated our results in Fig. 8, which demonstrates the capability of our INSP-Net in processing complex geometries encoded by INRs.

---

### Official Review · Reviewer_1Ys6 · 2022-07-11

**Rating:** 7
**Confidence:** 4
**Soundness:** 4 excellent
**Presentation:** 4 excellent
**Contribution:** 3 good

**Summary:**

The paper proposes a method to perform signal processing of a signal represented via a neural field directly, without the need to first sample the underlying MLP. To this end, the authors propose to express operators on top of the neural field as functions of the first- and higher-order derivatives of the MLP. Though the output is *not* a new, monolithic MLP parameterizing the output of the operator, but instead a linear combination of derivative MLPs, the output nevertheless retains the "resolution-invariant" property of the underlying neural field representation.


**Questions:**

Could the authors comment on the effective (not theoretical) receptive field of the operators that can be parameterized by the proposed approach? In particular, SIREN is known to be a highly local INR: the directions in weight space to change the colors of pixels even a small distance apart are basically orthogonal (increasingly so with higher frequencies). It would seem that this "local support" also somehow impacts the receptive field of a convolutional operator expressed via the gradients of the SIREN?

Do the authors have a way in mind in which they might lift the limitation that their approach requires a-priori fitting of all the neural fields? Note that I do not expect this to be the case, any insight is simply appreciated!

Minor comments:
- line 49: "are loaded from the INR being processing..." - this sentence does not make sense to me. Maybe being -> during?
- line 51: "even though we are not able to (perform) surgery"


**Limitations:**

The processing remains expensive, in that the output of the proposes INSP-Net is not a single neural field, but rather, a linear combination of derivative operators, each of which is expensive to query.

The present effort still requires recovering all neural fields of all the signals upfront.

The authors certainly discuss these limitations, but I would have expected a "limitations" section where these issues are made explicit.



**Strengths And Weaknesses:**

# Originality
The paper is original and the proposed approach novel. To the best of my knowledge, no prior work on directly computing convolutional operators on top of neural fields without sampling exists. The treatment of how to approximate any shift-equivariant and certain group-invariant kernels as the linear combination of first- and higher-order gradients of a neural field is similarly novel. The authors propose and demonstrate a set of applications, ranging from image processing to a simple convolutional neural network, that have also not been tackled in this way before. All in all, this paper is highly original and was joyful to read.

# Quality
The paper is of high quality. Figures are informative to a degree that a single pass through the paper already provides high-level understanding of the method. A detailed read-through yields cool insights, such as the derivation of universal approximation of any convolutional filter, though that is, as the authors remark, not a new result. The quality of the experiments is similarly high, with relevant experiments that show-case the use-case of the proposed method - in particular, I enjoyed the low-level image processing tasks and image inpainting.

# Clarity
The paper is very clear, especially the figures serve to lay out the key contributions of the paper.

# Significance
The practical significance of the proposed approach seems to be limited for now: First, it requires one to already have access to a set of neural fields fit to the signal at hand. Second, the cost of evaluating the resulting network can be rather prohibitive, as it requires evaluating the stack of all derivative networks to decode a single pixel.

However, I am not overly concerned with the practical significance of the approach. In my opinion, the key significance of this paper lies in educating the neural field community about the fact that operators on neural fields can be expressed as functions of the derivative networks. Further, I am convinced that this paper will inspire work that will push this idea further, maybe eventually to the point where we will be able to compute functions of neural fields directly in the weight space.

---

> ### Author Response · Authors · 2022-08-02
> **Response to Reviewer 1Ys6**
>
> We tremendously appreciate reviewer 1Ys6’s positive assessment and highlighting our contribution to the neural field community. In regards to your questions, see our responses below:
>
> **Q1. Could the authors comment on the effective (not theoretical) receptive field of the operators that can be parameterized by the proposed approach?**
>
> We appreciate this good question. Our insight is that even SIREN is a highly local INR, its derivatives can encode neighbor information. We can characterize the receptive field by looking back at the “equivalent” kernels in the discrete domain, where higher order differential operators are usually implemented by larger kernels. For example, first-order differential operators are often represented by a 3x3 kernel (e.g., sobel), while second-order differential operators can be approximated by 5x5 kernels. Likewise,  in the continuous domain, we can simply consider that higher order of adopted differential operators in INSP-Net induces a larger receptive field. Considering the effective receptive field defined in [1], we admit it remains an open question to directly compute the measure of impact between every pair of points because the computation of two points seems independent of each other in our INSP-Net.
>
> [1] Luo et al. Understanding the Effective Receptive Field in Deep Convolutional Neural Networks
>
>
> **Q2. How to lift the limitation that their approach requires a-priori fitting of all the neural fields?**
>
> Fitting an INR and processing an INR are two independent tasks. Our paper focuses on how to process an INR after acquiring it through a standard reconstruction pipeline. Avoiding a-priori fitting is beyond the scope of our discussion. Nevertheless, there indeed exists several effective solutions, such as latent code [1] and sparse coding [2], which acquire the representation in one-shot inference. Our INSP-Net is independent of the INR’s network architecture and is thus also generalizable to all those aforementioned methods.
>
> [1] Park et al. DeepSDF: Learning Continuous Signed Distance Functions for Shape Representation.
>
> [2] Wang et al. Neural Implicit Dictionary Learning via Mixture-of-Expert Training.
>
> **Q3. Suggestions on typos and a limitation section.**
>
> Thanks for these constructive comments. We have fixed all the grammar errors and added a limitation section in our revision which clarifies all the limitations mentioned by the reviewer.

---

> ### Author Response · Authors · 2022-08-06
> **Response to Reviewer 1Ys6**
>
> Dear Reviewer 1Ys6:
>
> We would like to thank you again for the favorable assessment of our work. We have put our two cents in your insightful open questions, and hope they are helpful. As a follow-up on our responses, we would like to kindly ask if you want to have further discussion as the author-reviewer interaction session is ending soon. We would be more than happy to provide more information or clarification.
>
> Any suggestions to improve the quality of this paper would be greatly appreciated. We sincerely wish to bring our theoretical grounded solution for direct signal processing on INR into the neural field community.
>
> Best,
>
> Authors of paper 988

---

> > ### Comment · Reviewer_1Ys6 · 2022-08-07
> > **Response**
> >
> > Thank you - I am happy with the changes you made. Minor comments:
> >
> > 1. I think the section should be called "Limitations." instead of "Limitation".
> > 2. I think in the Limitations section, you should clarify that you aren't addressing how to reconstruct / infer INRs, and that the present paper does not give any hint of how that might be achieved in a scalable manner.
> >
> > All in all, I remain positive about this paper, and will champion it the discussion with the other reviewers.

---

> > > ### Author Response · Authors · 2022-08-08
> > > **Response to Reviewer 1Ys6**
> > >
> > > We appreciate the reviewer's positive comments. We've made changes in our revised draft as suggested by the reviewer.

---

### Author Response · Authors · 2022-08-02
**General Response**


We thank all reviewers for their time and detailed comments. We are glad that all the reviewers acknowledged our strong motivation and novelty. We also noticed some of the reviewers are more or less concerned about our experimental results. Hereby, we would like to re-state the significance of our work.

INR has achieved ubiquitous success in signal compression [2], 3D reconstruction [1,3,4], and even scientific computing  [1,5,6]. Representing signals with a coordinate-based neural network enjoys promising properties, such as continuous representation and unlimited resolution [1]. These advocate INR potentially becoming the next universal data format for the future. However, being represented by implicit neural network weights, such signals cannot be manipulated or processed easily unless decoding INR into grid-based representations. The proposed framework is the **first** generic framework that achieves INR editing directly on the weight space and network structures, with a **theoretical guarantee**.

The goal of this paper is not to beat the state-of-the-art baselines on low-level vision or image classification tasks. Instead, our experiments are designed to demonstrate the effectiveness of leveraging differential operators to analytically process INRs. Although the experimental results do not demonstrate superiority in every aspect, they are still encouraging considering this is the first time that implicit fields can be processed and even learned on the weight space. To further demonstrate the practical significance, we also added experiments on geometry and audio processing.

According to reviewers’ suggestions, we have made these several changes (highlighted in blue) to our PDF, summarized as below :

1. We have added experiments in geometric processing (Sec. 5.3) and audio denoising (Fig. 14). This extends the application range of our work so that we can safely claim our method is for general-purpose signal processing.
2. We have added quantitative comparisons on a large dataset in Tab. 2 & Tab. 3.
3. We have added a limitation paragraph (Sec. 6) to clarify the existing downsides of our methods, which are left for future works.
4. We have carefully revised our manuscript correcting detail errors and adding missing references.
5. Due to page limit, we deferred the detailed introduction to INSP-ConvNet to the Appendix C, and added pseudocode for clarity.

We would appreciate it if all reviewers could please take a look and finalize their assessments of our work, hopefully more positively. We trust the reviewer and AC discussion would eventually lead to an informed and fair decision, and we thank everyone again for the valued efforts!

Best,

Paper 988 Authors

[1] Sitzmann et al. Implicit Neural Representations with Periodic Activation Functions

[2] Dupont et al. COIN: COmpression with Implicit Neural representations

[3] Park et al. DeepSDF: Learning Continuous Signed Distance Functions for Shape Representation

[4] Mildenhall et al. NeRF: Representing Scenes as Neural Radiance Fields for View Synthesis

[5] Li et al. Fourier Neural Operator for Parametric Partial Differential Equations

[6] Zhong et al. CryoDRGN: reconstruction of heterogeneous cryo-EM structures using neural networks

---

### Comment · Area_Chair_i8uP · 2022-08-07
**Discussion period**

Thank you to all the reviewers for the great effort in reviewing the paper and the authors for the responses.

As the author-reviewer discussion period is almost over, I want to ensure that reviewers have read the authors' responses and engage with the authors if needed.

If you haven't done this, could you please take a moment to read through the authors' responses, update the reviews to indicate that you have read the authors' responses, or communicate with the authors if needed? You can also share in private conversations with the reviewing team.

Please continue to share your thoughts. Thank you!

---

### Meta-Review · Area_Chair_i8uP · 2022-08-21

**Recommendation:** Accept
**Confidence:** Certain

**Metareview:**

The paper proposes a framework to perform signal processing tasks on a signal represented with an implicit neural representation directly in the representation space, without the need to instantiate the signal.

After the rebuttal period, all reviewers recommend acceptance.

In particular reviewer 1Yx6, an expert on the topic, finds the idea original, and the quality and clarity of the paper to be high. The reviewer finds that while for now the significance is limited (since working as proposed in the representation space is computationally prophibitavely expensive), this is not a major issue, since the the paper is likely to inspire work that will push the idea further.
Reviesers edqT and qGk1 also liked the general idea of the paper and find the proposed method to directly perform operations on the representation space of implicit neural representations to be novel and interesting

Reviewer Uh51 initially identified a few issues regarding experimental design, the validation, and on the included literature. The main concern regarding the experimental design of the reviewer was that the paper focuses on images (and not signal processing tasks more broadly), which I don't consider a shortcoming, due to the importance of image processing tasks. The concerns on the validation issues have been addressed as well, and the reviewer raised their score.

I recommend acceptance of the paper.


**Award:**

No

---

### Decision · Program_Chairs · 2022-09-14

Accept